# Unsupervised logic-based mechanism inference for network-driven biological processes

**Martina Prugger**[1,2], **Lukas Einkemmer**[3], **Samantha P. Beik**[2], **Perry T. Wasdin**[2], **Leonard A. Harris**[2,4,5,6], **Carlos F. Lopez**[2,7] *

1 Department of Biochemistry, University of Innsbruck, Innsbruck, Austria, 2 Department of Biochemistry, Vanderbilt University School of Medicine, Nashville, Tennessee, United States of America, 3 Department of Mathematics, University of Innsbruck, Innsbruck, Austria, 4 Department of Biomedical Engineering, University of Arkansas, Fayetteville, Arkansas, United States of America, 5 Interdisciplinary Graduate Program in Cell and Molecular Biology, University of Arkansas, Fayetteville, Arkansas, United States of America, 6 Cancer Biology Program, Winthrop P. Rockefeller Cancer Institute, University of Arkansas for Medical Sciences, Little Rock, Arkansas, United States of America, 7 Department of Biomedical Informatics, Vanderbilt University Medical Center, Nashville, Tennessee, United States of America

* c.lopez@vanderbilt.edu

**Data Availability Statement:** All code and source files are available from GitHub at our lab URL: https://github.com/LoLab-VU/Boolean_rules_creator.

## Abstract

Modern analytical techniques enable researchers to collect data about cellular states, before and after perturbations. These states can be characterized using analytical techniques, but the inference of regulatory interactions that explain and predict changes in these states remains a challenge. Here we present a generalizable, unsupervised approach to generate parameter-free, logic-based models of cellular processes, described by multiple discrete states. Our algorithm employs a Hamming-distance based approach to formulate, test, and identify optimized logic rules that link two states. Our approach comprises two steps. First, a model with no prior knowledge except for the mapping between initial and attractor states is built. We then employ biological constraints to improve model fidelity. Our algorithm automatically recovers the relevant dynamics for the explored models and recapitulates key aspects of the biochemical species concentration dynamics in the original model. We present the advantages and limitations of our work and discuss how our approach could be used to infer logic-based mechanisms of signaling, gene-regulatory, or other input-output processes describable by the Boolean formalism.

## Author summary

Mechanisms of biological processes that explain and predict biological behaviors continue to be challenging to attain. In this context, logic-based models with few parameters can be formulated to describe experimental data. However, constructing such networks based on the available evidence is often done in an ad-hoc, error-prone manner that reflects the bias of the modeler. Here we present an algorithm that infers Boolean logic models from mappings of initial states to steady states, from available experimental data, and without

**Funding:** This work was supported by the National Science Foundation (NSF) [MCB 1411482] and NSF CAREER Award [MCB 1942255] to CFL; the National Institutes of Health (NIH) [U54-CA217450 and U01-CA215845] to CFL; the National Library of Medicine (NLM) Vanderbilt Biomedical Informatics Training Program (NLM 5T15-LM007450-14) to LAH; and the NIH/National Cancer Institute (NCI) Transition Career Development Award to Promote Diversity (K22-CA237857-01A1) to LAH. The funders had no role in study design, data collection and analysis, decision to publish, or preparation of the manuscript.

**Competing interests:** The authors have declared that no competing interests exist.

human supervision. Moreover, our methodology enables users to incorporate additional biological information (expert knowledge) to further refine Boolean models of cellular processes.

## Introduction

A mechanistic understanding of dynamic cellular processes is at the core of multiple areas of research including molecular cell biology, physiology, biophysics, and bioengineering [1–6]. Although analytical tools have improved the breadth and depth with which intra- or extra-cellular biochemical processes are explored [7–9], the vast majority of available data is limited to experiments that probe cue-response relationships with a specified set of inputs and outputs. Although significant efforts have been devoted to understand how biochemical interactions link these inputs and outputs, the formulation of mechanistic hypotheses remains a challenging problem, yet essential to explain and predict cellular responses to perturbations [10–15].

The Boolean logic formalism, introduced by Kauffmann in 1969 [16], is a simple and powerful approach to describe gene-regulatory networks, signaling networks, metabolic networks, and many others [17, 18]. In this representation, each node in a network corresponds to a gene or gene-product while each edge corresponds to a Boolean rule or set of rules that describes the interaction between nodes. The system can evolve for a number of discrete steps, where the state of each node (one or zero) is determined by evaluating its associated logic rules at each step. The system is typically evolved for a number of steps using a Markov-chain process until a steady state (aka attractor state) is achieved [19]. In biological networks, the nodes can have multiple biological meanings, e.g., genes or proteins, while the edges represent interactions between the nodes. The transition of one state of nodes to another can be interpreted as the fully executed chemical reaction that leads to another state of the system. Attractors in such a network are states of the system that can no longer transition into another state and are interpreted as steady states. Multiple attractors, for example, can be interpreted as multiple phenotypes of the same cancer cells in a heterogeneous cell population. Translating continuous data into Boolean states is typically done by setting a threshold where everything below is set to zero and everything above is set to one. A discussion about how to deal with data where such a threshold is not obvious can be found, e.g., in [15]. Boolean representations of biochemical reaction networks have yielded important biochemical insights [20–25] and offer a parameter-free alternative to other formalisms where exact parameters may be difficult or impossible to acquire [26, 27].

Despite the mathematical simplicity of Boolean logic based biochemical networks, the interaction rules that dictate the dynamics cannot be directly obtained from either experimental data or curated interaction databases [28–30]. For this reason, logic rules enumeration, which comprise a specific mechanism of action, remains a central challenge in Boolean logic modeling. This problem can be found in all areas of biology as well as other fields of research (e.g. electrical engineering, natural language [31, 32], and even search engines employ Boolean techniques) where Boolean modeling is employed. Therefore, our goal was to develop a rigorous methodology to automatically generate Boolean rules, given input and output states, and generate mechanistic hypothesis to link network states within biological constraints without human involvement beyond initial setup (i.e. unsupervised).

Assembling a Boolean-logic based model from experimental data is typically done by hand, using informal reasoning for both network structure and Boolean rules inference even for a large number of species [33, 34]. A common approach to bypass this cumbersome task

is to translate a rule-based network into Boolean rules [35, 36]. Tools such as CellDesigner [37] offer the possibility to visually construct such models using interaction graphs that can then be translated into their Boolean counterpart via automated tools, e.g., [38]. Boolean models can also be constructed using the data representation in their algebraic equations form, rather than identifying the logic relations directly [39]. Data that translate into Boolean rules can be taken from various sources including time series data, as used in [40–42] and analyzed by tools such as BoolNet [43]. The attractors obtained from these formalisms are therefore model predictions, given a specified mechanism rather than the inverse problem of formulating a mechanism for a set of observables [40, 42]. Although it is desirable to preserve experimentally observed attractors, there is no guarantee for such models that a given initial state necessarily evolves towards the correct steady states [44]. Enforcing such constraints manually is often a tedious and error-prone process. A proposed way of preserving the steady state structure for synchronous updating schemes is proposed in, e.g., [45]. The chosen updating scheme for model evolution, however, can significantly affect the interpretation of model dynamics. For example, synchronous updating schemes may yield network dynamics with no clear biological interpretation [46–49]. By contrast, sequential node updating schemes, such as General Asynchronous, can provide a mechanism with better biochemical correlation [22, 50–53].

In this work, we address the problems of mechanism inference in biological processes where input states and attractors are known but the mechanism is unknown. The proposed algorithm constructs both candidate network structure and the corresponding Boolean rules in an unsupervised manner. That is, the algorithm iteratively performs model selection without human intervention. Our method guarantees that the selected initial states reach their designated steady states, that no spurious steady states are introduced, and that the network logic inferred is compatible with the biologically relevant asynchronous updating [44]. In addition, experimentally-observed probability distributions from one initial state to multiple attractor states are preserved by our algorithm—often a biologically important observation. Our algorithm can thus be used for hypothesis exploration, model identification, and mechanism exploration *in silico* in the context of complex experimental data.

Our proposed method explores the Boolean mechanism or the network dynamics and phrases the resulting network in the form of a Boolean rule set. We also want to point out that in this paper, we are creating a network based on the state transition graphs and formulating them into Boolean rules.

## Methods

In this section, we introduce how the algorithm works using enzyme-substrate dynamics. To demonstrate the working of the algorithm, we start with a network in which pathways only go forward towards the attractors. Then we will extend the method towards all possible network connections and, in the end of the section, we suggest how to make a more informed network choice. The main idea of the proposed algorithm is shown in Fig 1. As input, a mapping between each initial state and the corresponding steady states is given. In asynchronous updating, as we consider here, the state of only one species can be changed per step. This means that the Hamming distance (i.e., the number of bits that differ between the states) of all states that are reachable in the next step is equal to one [54]. We exploit this knowledge to construct paths from each initial value to the reachable steady states, while avoiding paths that lead to incorrect results. This allows us to generate (in general many) candidate networks that satisfy precisely the prescribed mapping. The probability distribution of the steady states can also be specified. This is then used, within a genetic optimization algorithm, to select models which

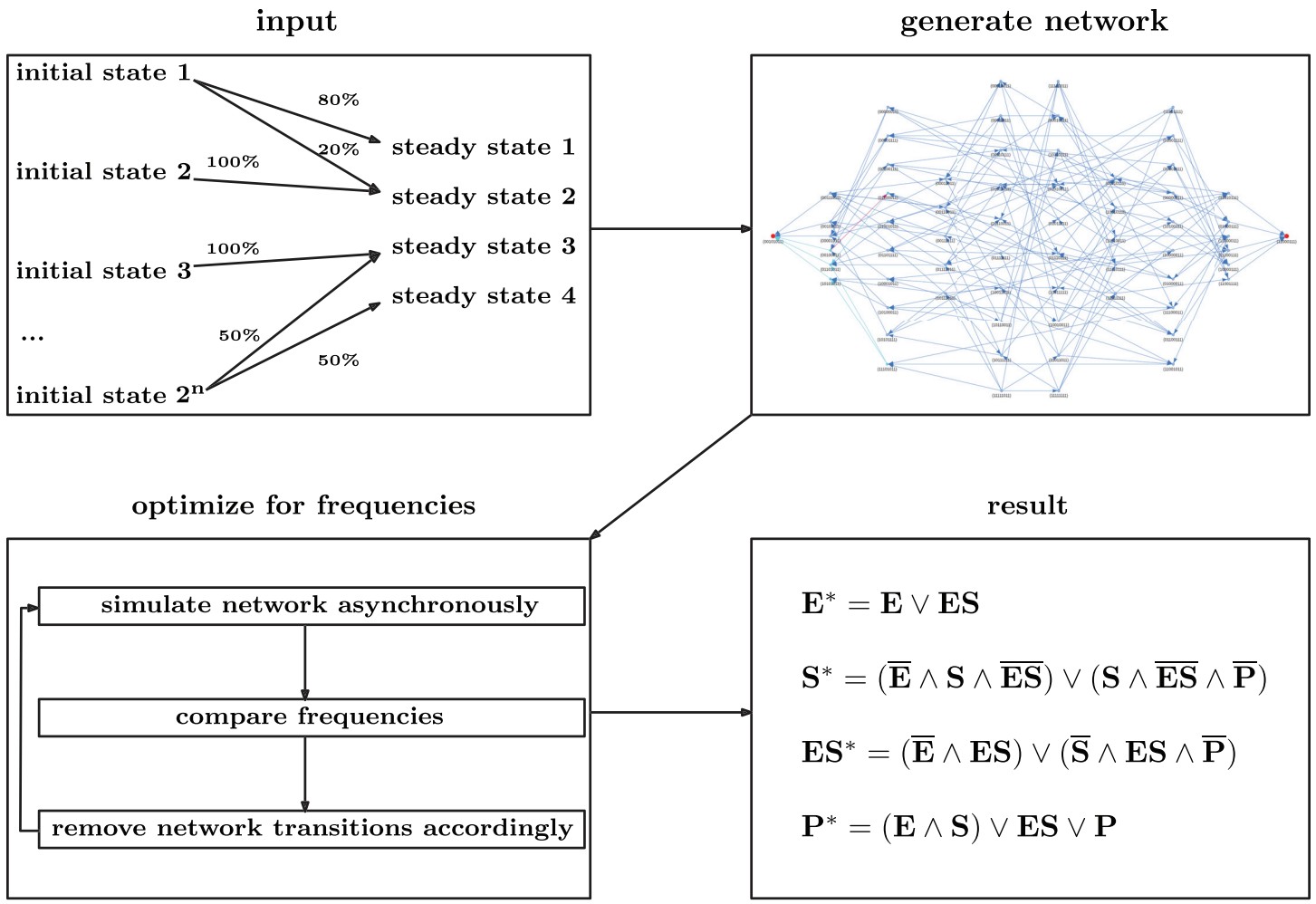

**Fig 1. Schematic depiction of our workflow.** Our input data is of the form initial state and corresponding steady states. If the same initial state is observed to end up in multiple steady states, a probability to reach each steady state can be prescribed as well. From the initial state to attractor relationship, a network is created using the state transition graph, taking into account every possible connection allowed by an asynchronous updating scheme. We then simulate the network and compare the resulting probabilities to the specified measurement data. If necessary, we remove transitions from the network to achieve a better match between the probabilities of the resulting network and the the experimental data. The result is a system of Boolean rules that describes the network dynamics.

show the same dynamics. At this point, expert knowledge on the network (such as on which species a given rule depends on or specific transitions that should be included) can be incorporated as well. A number of good candidate models are then selected and the corresponding Boolean rules are generated. The algorithm automatically simplifies these results using symbolic manipulation. The algorithm proposed is described in the following sections for two examples: an Enzyme-Substrate kinetics model and an established Epithelial to Mesenchymal Transitions (EMT) epithelial mouse cancer cell metastasis [55, 56].

The basic algorithm is explained in some detail for the Enzyme-Substrate kinetics reaction mechanism which is facilitated by the smaller size of that particular problem. However, all the steps in the algorithm (except for the problem specific expert guidance that can be used) have been fully automated and are part of a parallelized hybrid Python/C++ code. Thus, the creation of the Boolean rules is done fully automatically and the detailed enumeration of some of those steps for the Enzyme-Substrate kinetics reaction mechanism problem are only provided as examples to gain a better understanding of the algorithm. For both examples we show how

to incorporate expert knowledge into our method. This by necessity is problem dependent and thus different approaches, that should generalize well to many other problems, are explored.

We note however, that our approach generates candidate networks based on the transition graphs and formulates Boolean rules from said graphs. Therefore, network representations should not be confused with interaction or regulatory networks, where network edges correspond to biochemical interactions. It follows that a Boolean pathway describes the possible paths that a simulation can take to reach the steady state in the network, and has no connection to e.g. signaling pathways. Finally, a Boolean rule does not necessarily possess a chemical mechanism interpretation. Our method explores the Boolean mechanism or the network dynamics and phrases the resulting network in the form of a Boolean rule set. The (bio)chemical interpretation of these rules may not be obvious from a biological perspective and further exploration and simplification may be necessary to interpret the biological meaning of a given logic rule.

## Network inference for Enzyme-Substrate dynamics

We employ an enzyme-substrate reaction system to demonstrate the details of our approach. In this representation $E$ is the enzyme, $S$ is the substrate, and $P$ is the resulting product. The enzyme can bind to the substrate into the complex $ES$ via a specific rate $k_f$ and break up into the two species via the rate $k_r$ or catalyze the substrate-to-product reaction, resulting in free enzyme and product according to the chemical equation

$$E + S \underset{k_r}{\overset{k_f}{\rightleftharpoons}} ES, \qquad ES \overset{k_c}{\rightarrow} E + P. \tag{1}$$

Mathematically, this results in a system of ordinary differential equations (ODEs) with species concentration $E$, $S$, $ES$, and $P$ as well as the three parameters $k_f$, $k_r$, and $k_c$.

Our goal is to model the corresponding dynamics using a Boolean network. Boolean networks assume that the species are either present (1) or absent (0), i.e. $E, S, ES, P \in \{0, 1\}$, and that all reactions are equally likely, i.e. all rate constants are equal to 1. To match this we will also make the assumption that $k_f = k_r = k_c = 1$ in our enzyme-substrate reaction. For the concentrations we will start with either 1 or 0 for each species, but the concentration is allowed to take on fractional values as the reaction dynamics evolve. The results of such a simulation are shown in Fig 2. The initial value to steady state mapping so obtained will be used in this section to automatically construct a Boolean network using the proposed algorithm. The goal of this Boolean network is to recover the dynamics of the ODE simulation. Let us note that in the Boolean model the values of the species are necessarily either 1 or 0. We will, however, interpret the average of the stochastic asynchronous update as a (relative) concentration similar to the one found in the ODE model.

For this particular system, we have four species each of which can take on two conditions, for a total of $2^4 = 16$ possible states of the system, namely:

$$(E, S, ES, P) \in \{(0, 0, 0, 0), (0, 0, 0, 1), (0, 0, 1, 0), (0, 0, 1, 1), (0, 1, 0, 0), (0, 1, 0, 1),$$
$$(0, 1, 1, 0), (0, 1, 1, 1), (1, 0, 0, 0), (1, 0, 0, 1), (1, 0, 1, 0), (1, 0, 1, 1),$$
$$(1, 1, 0, 0), (1, 1, 0, 1), (1, 1, 1, 0), (1, 1, 1, 1)\}.$$

For consistency, each tuple represents the species in the order as shown (i.e. the first entry is the $E$ state, the second entry is the $S$ state, the third is the $ES$ state, and the fourth entry is the $P$ state). Once all states have been defined we can analyze the states for biochemical significance. For example, state $(0, 0, 0, 0)$ signifies that no species are present in the system and

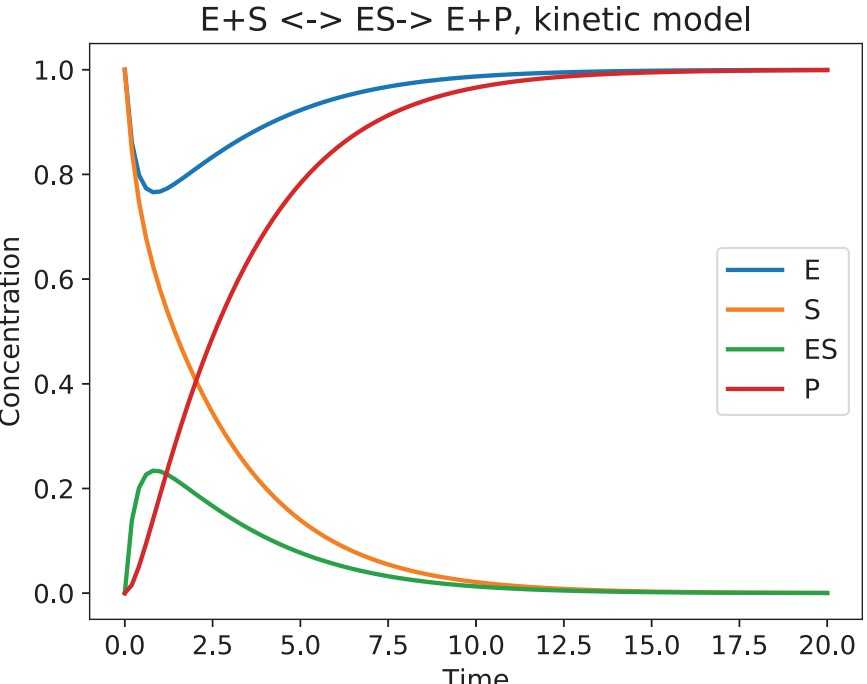

**Fig 2. Reaction kinetics of an enzyme-substrate system with rate constants $k_f = k_r = k_c = 1$ and initial concentrations $E = S = 1$ and $ES = P = 0$.** The simulation is the solution of the underlying ordinary differential equation, and the concentrations of the species are therefore still $\in \mathbb{R}_{[0,1]}$. The dynamics depicted in this graph is considered to be the underlying truth that our algorithm tries to recreate with an automatically generated Boolean logic network.

therefore no chemical reactions can occur. States (1, 0, 0, 0), (0, 1, 0, 0), and (0, 0, 0, 1) similarly have only one of enzyme, substrate, or product present and similarly no chemical reactions can take place. Absence of reactions is also seen in the state (0, 1, 0, 1), since substrate and product do not interact with each other. Removing those states form further considerations leads to a network that treats them as a steady state which can not be accessed by any other state. All other states, however, converge towards the attractor state (1, 0, 0, 1). The first step of our introduced method is to match all initial states to their according steady state represented in Table 1.

**Table 1. Mapping of the initial states to their corresponding steady state.** By mapping states to themselves, they create a steady state for the network that can not be accessed by any other state. Note, that in principle, initial states can converge towards multiple different steady states. This behavior is captured easily by just adding these states to all of the corresponding steady state lists.

| Steady State | Initial States |
|---|---|
| (0, 0, 0, 0) | (0, 0, 0, 0) |
| (0, 0, 0, 1) | (0, 0, 0, 1) |
| (0, 1, 0, 0) | (0, 1, 0, 0) |
| (0, 1, 0, 1) | (0, 1, 0, 1) |
| (1, 0, 0, 0) | (1, 0, 0, 0) |
| (1, 0, 0, 1) | (0, 0, 1, 0), (0, 0, 1, 1), (0, 1, 1, 0), (0, 1, 1, 1), (1, 0, 0, 1), (1, 0, 1, 0), (1, 0, 1, 1), (1, 1, 0, 0), (1, 1, 0, 1), (1, 1, 1, 0), (1, 1, 1, 1) |

**Table 2. Sorting of the initial states according to their Hamming distance from the steady state (1, 0, 0, 1).**

| Hamming distance from steady state (1, 0, 0, 1) | Initial States |
|---|---|
| 1 | (1, 0, 1, 1), (1, 1, 0, 1) |
| 2 | (0, 0, 1, 1), (1, 0, 1, 0), (1, 1, 0, 0), (1, 1, 1, 1) |
| 3 | (0, 0, 1, 0), (0, 1, 1, 1), (1, 1, 1, 0) |
| 4 | (0, 1, 1, 0) |

In the next step, for each attractor, the initial states are sorted according to their Hamming distance from the steady state. The sorting for steady state (1, 0, 0, 1) is listed in Table 2.

Based on the asynchronous updating scheme, a state with Hamming distance $n$ will require at least $n$ updates to reach the attractor. We then identify the transitions necessary to build a pathway for each Hamming distance level.

We use this information to create a transition map for each species that contains the necessary transformations to reach a given attractor. We achieve this by working our way backwards from each attractor. For example, for a level one (i.e. Hamming distance = 1) transition, the state (1, 0, 1, 1) needs to flip the third bit (the bit for $ES$) to reach the attractor (1, 0, 0, 1). Similarly the state (1, 1, 0, 1) needs to flip the second bit (the bit for $S$) to reach the attractor (1, 0, 0, 1). Therefore, the transition lists for $S$ and $ES$ will be updated with the states (1, 1, 0, 1), and (1, 0, 1, 1) respectively. We do the same for level 2 (i.e. states with Hamming distance = 2), as well as for all other levels and extend the lists accordingly. The full sorting can be found in Table 3. Note, that for a system with multiple attractors, each attractor gets a similarly created list.

A graphic representation for the corresponding pathways can be found in the transition graphs S1 and S2 Figs in the supplementary material. This list includes all the necessary transitions for each species to reach a given attractor. In a system with multiple steady states, this procedure has to be performed for each attractor. With the transition list, we can then infer the Boolean rules to update each species according to its associated transition list using the following steps:

1. Each binary number is matched as a Boolean expression. For the enzyme-substrate system, e.g., the state (0, 0, 1, 1) is matched by the Boolean expression that only returns *True*, if the state is E = 0 (or *not* E), S = 0 (or *not* S), ES = 1, and P = 1. Since every other state returns *False*, this is equivalent to connecting each element of the expression via the logical *AND* operator, i.e.: (*not* E *AND not* S *AND* ES *AND* P). From now on, we will continue by using the mathematical notation, where *not* X is written as $\overline{X}$, and *AND* is written as $\wedge$, i.e., $(\overline{E} \wedge \overline{S} \wedge ES \wedge P)$

In the first step, we translate every state from Table 3 into its corresponding Boolean expression.

**Table 3. List of transitions for each species that make up the network pathways sorted by their Hamming distance $d$ to the steady state (1, 0, 0, 1).** Note, that a state can reach the steady state in multiple ways. It is therefore possible to have the same initial assigned to multiple species.

| species | list for $d$ = 1 | list for $d$ = 2 | list for $d$ = 3 | list for $d$ = 4 |
|---|---|---|---|---|
| E | | (0, 0, 1, 1) | (0, 0, 1, 0), (0, 1, 1, 1) | (0, 1, 1, 0) |
| S | (1, 1, 0, 1) | (1, 1, 1, 1) | (0, 1, 1, 1), (1, 1, 1, 0) | (0, 1, 1, 0) |
| ES | (1, 0, 1, 1) | (1, 1, 1, 1) | (1, 1, 1, 0) | |
| P | | (1, 0, 1, 0), (1, 1, 0, 0) | (0, 0, 1, 0), (1, 1, 1, 0) | (0, 1, 1, 0) |

2. Let us now look at the first row of Table 3, namely the transitions for the enzyme, which is *True* for the four Boolean expressions ($\overline{E} \wedge \overline{S} \wedge ES \wedge P$), ($\overline{E} \wedge \overline{S} \wedge ES \wedge \overline{P}$), ($\overline{E} \wedge S \wedge ES \wedge P$), and ($\overline{E} \wedge S \wedge ES \wedge \overline{P}$). This means, that *E* is set to *True*, if any of these expressions is *True*. We therefore combine the four expressions via the logical *OR* operator notated by $\vee$.

3. By construction of the network, the enzyme E is supposed to be flipped, if the expression from step 2 is *True* and not flipped in any other case. This means, that we need an if-condition of the form *if*(expression from step 2 is *True*): change the current state of E, *else*: keep E as it was. This can be achieved by the logical exclusive *OR* operator, for short *XOR* ( $\veebar$ ), which returns *True* if and only if one of the conditions of the input is *True*. Contrary to the *OR* operator, *XOR* returns *False*, if both input conditions are met. If we therefore couple the expression from step 2 with E using the *XOR* operator in this way (expression from Point 2 *XOR* E), we have the following possibilities:

- E = 0 and none of the states from Table 3 is met → the input for the *XOR* operator is (*False* $\veebar$ *False*), and E stays E = 0.

- E = 0 and one of the states is met → (*True* $\veebar$ *False*), and E gets updated to E = 1.

- E = 1 and none of the states is met → (*False* $\veebar$ *True*), and E stays E = 1.

- E = 1, and one of the states is met → (*True* $\veebar$ *True*), and E gets updated to E = 0.

Therefore, the updating rule for E can be written as:

$$((\overline{E} \wedge \overline{S} \wedge ES \wedge P) \vee (\overline{E} \wedge \overline{S} \wedge ES \wedge \overline{P}) \vee (\overline{E} \wedge S \wedge ES \wedge P) \vee (\overline{E} \wedge S \wedge ES \wedge \overline{P})) \veebar E$$

Using Boolean algebra, this expression can be further simplified to

$$E \vee ES$$

The same procedure for the other species results in the following rules

rule for *S*:

$$((E \wedge S \wedge \overline{ES} \wedge P) \vee (E \wedge S \wedge ES \wedge P) \vee (\overline{E} \wedge S \wedge ES \wedge P) \vee (E \wedge S \wedge ES \wedge \overline{P})$$

$$\vee (\overline{E} \wedge S \wedge ES \wedge \overline{P})) \veebar S = (\overline{E} \wedge S \wedge \overline{ES}) \vee (S \wedge \overline{ES} \wedge \overline{P})$$

rule for *ES*:

$$((E \wedge \overline{S} \wedge ES \wedge P) \vee (E \wedge S \wedge ES \wedge P) \vee (E \wedge S \wedge ES \wedge \overline{P})) \veebar ES = (\overline{E} \wedge ES) \vee (\overline{S} \wedge ES \wedge \overline{P})$$

rule for *P*:

$$((E \wedge \overline{S} \wedge ES \wedge \overline{P}) \vee (E \wedge S \wedge \overline{ES} \wedge \overline{P}) \vee (\overline{E} \wedge \overline{S} \wedge ES \wedge \overline{P}) \vee (E \wedge S \wedge ES \wedge \overline{P})$$

$$\vee (\overline{E} \wedge S \wedge ES \wedge \overline{P})) \veebar P = (E \wedge S) \vee ES \vee P$$

Let us now try to interpret these rules from a chemical kinetics point of view: We know, that the steady state for this example has the enzyme turned on, i.e., $E = 1$. According to the rule $E^* = E \vee ES$, *E* is either already on or it is turned on by *ES*. *S* is turned off in the steady state, and according to the rule, it gets turned off, if *E* and *ES* are off at the same time, or if *ES* is off and *P* is on at the same time. *ES* is turned off, if *E* is turned off or *S* and *P* are turned off at the same time, while *P* either is already on, *ES* is on, or *E* and *S* are on at the same time.

**Dynamic behavior of the forward-only (ES-F) network.** The described way to obtain Table 3 only includes the transitions in one direction (from initial state to steady state) and therefore qualifies as "forward-only" network that does not allow for transitions that step backwards away from the steady state. The automatically created network evaluating forward-only interactions results in the following set of update rules

$$E^* = E \vee ES$$

$$S^* = (\overline{E} \wedge S \wedge \overline{ES}) \vee (S \wedge \overline{ES} \wedge \overline{P})$$

$$ES^* = (\overline{E} \wedge ES) \vee (\overline{S} \wedge ES \wedge \overline{P})$$

$$P^* = (E \wedge S) \vee ES \vee P.$$

The generated network structure of the forward network is depicted in Fig 3 top left. Using the General Asynchronous updating scheme [50, 51, 53] for the initial state (1, 1, 0, 0) for 100 simulations yields the result shown in Fig 3 on the top right. The depicted initial condition is the same as we have used for the kinetic simulation in Fig 2 to compare and judge the quality of our result.

The kinetic model in Fig 2 depicts concentrations of the species in the system, i.e, $E, S, ES, P \in \mathbb{R}_{\geq 0}$. Since this is not possible for a Boolean simulation, where $E, S, ES, P \in \{0, 1\}$, to capture the overall dynamics of the network, multiple simulations have to be performed. The random nature underlying the General Asynchronous updating scheme results in different pathways taken by each simulation. Looking for each species at the fraction of how many simulations are 0 or 1 at each simulation step allows us to capture dynamics similar to the kinetic description.

As we can see, the correct steady state is achieved. However, in these dynamics the enzyme never gets bound to the substrate, but substrate is directly converted into the product. The middle part of the enzyme-substrate kinetics is thus omitted. This is clearly not the desired dynamics.

A look at the graphic representation of the network as depicted in Fig 3, top left, gives us further insight into this problem. We can see that by our basic construction, we only allow the direct "forward" pathway for the network: The state (1, 0, 0, 0), i.e., the state where substrate is consumed has not been included into the network and therefore, the only possibility of our initial state to change is the creation of the product into the state (1, 1, 0, 1). This state has also no other path than directly go to the final state (1, 0, 0, 1), i.e., consume the substrate. However, no circulation or other dynamics are allowed in this network. Michaelis-Menten kinetics, however, are only a simple example that already demonstrates, that circulation within the pathways are an important biological factor of networks.

We therefore propose to extend our method to include the backward pathways into the network as well.

**Backward dynamic paths to enable dynamic loops (ES-B).** The list from Table 3 only accounts for transitions in the forward direction towards the attractor. By extending the corresponding lists to include the backward transitions, we generate a new rule-set that extends to a new network including all possible backward pathways as well. Note, however, that we can only include backward pathways starting with level 1 and higher. Including a backward path for level 0, i.e., the attractor, means that the simulation can leave this state and it therefore would be no longer be a steady state.

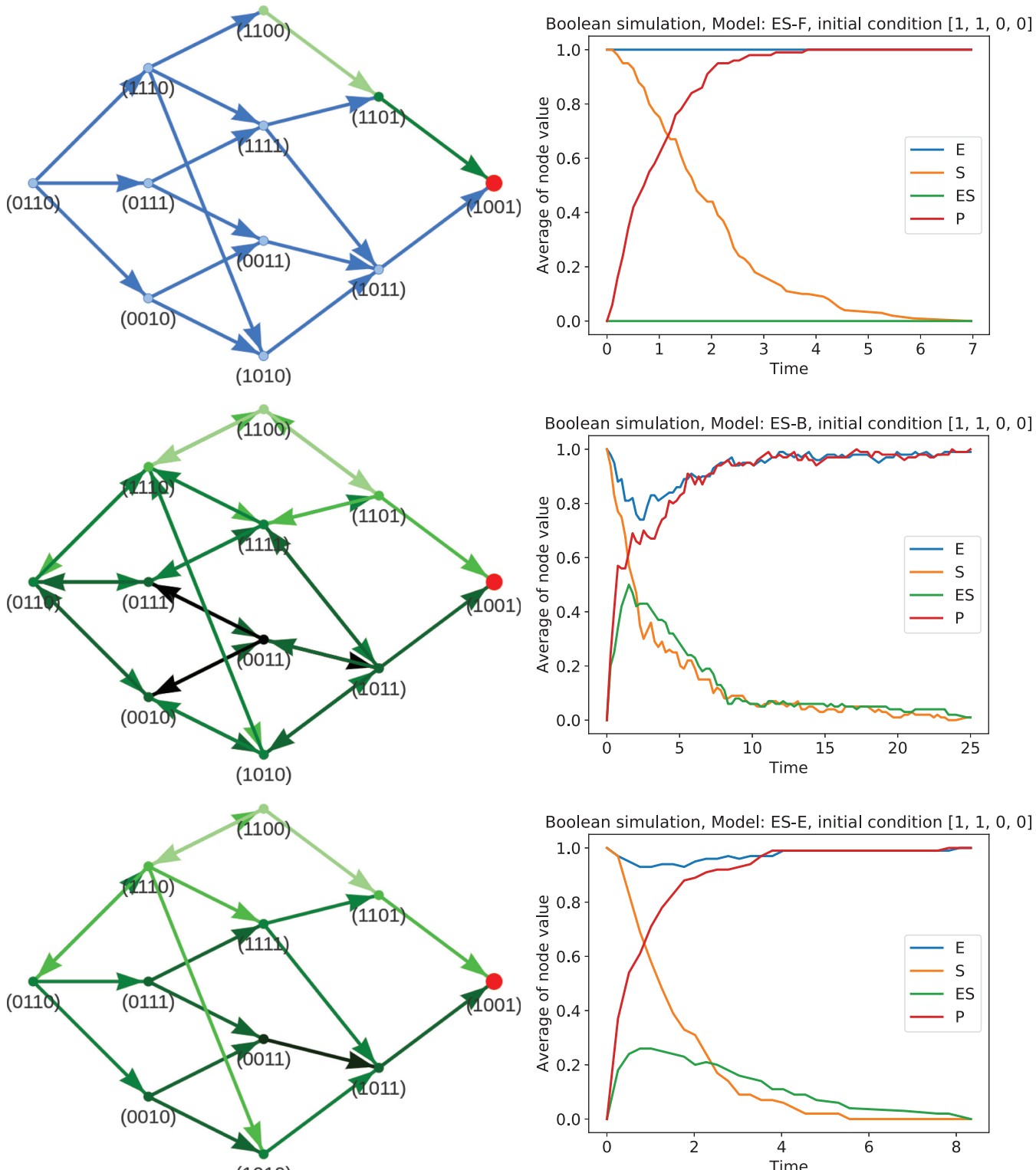

**Fig 3. State transition graphs for the Boolean networks created by our proposed rules generator (left) for the enzyme-substrate mechanics and their resulting dynamics after 100 asynchronous updating simulations for the initial state (*E, S, ES, P*) = (1, 1, 0, 0) (right).** The first row depicts the resulting network from the forward-only pathways network (ES-F), the second row the network including backward paths (ES-B), and the third row the network including expert knowledge (ES-E). For the network representation, the pathway for state (1, 1, 0, 0) to reach the steady state is highlighted in green. The darker the arrow, the more simulation steps are necessary to reach this particular transition. For the dynamics plots (right), the fraction of 100 simulations of the asynchronous updating scheme that are on/off is shown on the *y*-axis. The *x*-axis shows the number of simulation steps taken, representing the number of steps necessary for all 100 simulations to reach the steady state.

Using the extended transition sets and the same translation into updating rules, we obtain

$$
\begin{aligned}
E^* &= & (E \wedge \overline{ES}) \vee (\overline{E} \wedge ES) \\
S^* &= & (\overline{S} \wedge ES) \vee (\overline{E} \wedge S \wedge \overline{ES}) \vee (S \wedge \overline{ES} \wedge \overline{P}) \\
ES^* &= & (\overline{E} \wedge ES) \vee (E \wedge S \wedge \overline{ES}) \vee (\overline{S} \wedge ES \wedge \overline{P}) \\
P^* &= & (ES \wedge \overline{P}) \vee (E \wedge S \wedge \overline{P}) \vee (\overline{E} \wedge \overline{ES} \wedge P) \vee (\overline{S} \wedge \overline{ES} \wedge P).
\end{aligned}
$$

The network structure and simulation results for this rule-set and the initial state $(1, 1, 0, 0)$ is depicted in the middle row of Fig 3 on the left.

The network depiction demonstrates, that all paths in the network are now enabled. I.e., the state $(1, 1, 0, 0)$ can also transition into $(1, 1, 1, 0)$, which means that the product $ES$ is generated. We observe, that the simulation now both consumes enzyme and creates the complex $ES$ before the attractor is reached. The dynamics, depicted in the middle row of Fig 3 on the right, match the ground truth from Fig 2 well. We can also see, that it takes noticeably longer for all simulations to reach their overall attractor. This makes sense, since backward paths also enable simulations to loiter in loops.

Reduction of logic-rule search space with experimental data (ES-E). Both networks described above are created by automatically mapping initial states to their corresponding attractor without any additional knowledge. Due to the construction of our method, however, it is straightforward to include expert knowledge into the dynamics as well.

Let us for example look back at the construction of our first network. We have noted that in this case we omit the pathway for the creation of the complex $ES$. We are, however, aware that this part is a necessary step of the dynamics. In this example, we therefore propose to start with the forward-pathway network and add the transitions for $(1, 1, 0, 0)$ to $(1, 1, 1, 0)$, as well as the resulting consumption of $E$, namely the transition from $(1, 1, 1, 0)$ to $(0, 1, 1, 0)$ to the corresponding transition list.

The resulting ruleset is

$$
\begin{aligned}
E^* &= & (E \wedge P) \vee (E \wedge \overline{ES}) \vee (\overline{E} \wedge ES) \vee (E \wedge \overline{S}) \\
S^* &= & (\overline{E} \wedge S \wedge \overline{ES}) \vee (S \wedge \overline{ES} \wedge \overline{P}) \\
ES^* &= & (\overline{E} \wedge ES) \vee (\overline{S} \wedge ES \wedge \overline{P}) \vee (E \wedge S \wedge \overline{ES} \wedge \overline{P}) \\
P^* &= & (E \wedge S) \vee ES \vee P
\end{aligned}
$$

and in Fig 3 bottom, we see the resulting network (left) and the corresponding simulation for the initial state $(1, 1, 0, 0)$ (right). Since we only added the absolute minimum necessary to create $ES$, most of the loops from the backward pathways model are omitted and the simulation reaches the steady state in a similar time frame as the simulation with the forward pathways only while also capturing some of the dynamics of the $ES$ creation and $E$ consumption.

Note, that in this case we manually added transitions to the network we judged feasible. We also provide the option to exclude transitions that the user is certain are biologically unfeasible.

Our implementation enables the user to start with either the forward-path network, or the full backward path including network from which transitions can be added or removed as seen fit. Note, however, that not all removals are valid to keep the network dynamics: adding and/or removing random transitions could result in the following problems:

1. adding a transition that leads directly away from the attractor will result in a loss of this attractor as a steady state

2. adding a transition could create a pathway to the wrong attractor

3. removing a transition could make it impossible for a state to reach its attractor

Since the full backward path network includes all possible pathways between all nodes in the network, for an unknown process, we recommend to start with the full backward path network and start strategically removing transitions from there. This way, we can be certain that the necessary network connections are present, while we only need to assure that point 3. of the list is not violated. Note, however, that due to the many loops that are created in this network, more steps are required by the asynchronous updating scheme before equilibrium is achieved.

## Results

### Application to an established model: Epithelial to Mesenchymal Transition (EMT) in cancer cell metastasis

To demonstrate our mechanism inference approach in a real-world system, we infer the Boolean logic mechanism for the EMT transition observed in [55, 56]. The ruleset for the reference EMT model is:

$$
\begin{aligned}
\text{NICD}^* &= \text{Notch} \wedge \overline{\text{TP63\_TP73}} \wedge \overline{\text{TP53}} \\
\text{Notch}^* &= \text{ECM} \wedge \overline{\text{miRNA}} \\
\text{TP53}^* &= (\text{DNAdam} \vee \text{NICD} \vee \text{miRNA}) \wedge \overline{\text{EMTreg}} \wedge \overline{\text{TP63\_TP73}} \\
\text{TP63\_TP73}^* &= \text{DNAdam} \wedge \overline{\text{miRNA}} \wedge \overline{\text{NICD}} \wedge \overline{\text{TP53}} \\
\text{miRNA}^* &= (\text{TP53} \vee \text{TP63\_TP73}) \wedge \overline{\text{EMTreg}} \\
\text{EMTreg}^* &= \text{NICD} \wedge \overline{\text{miRNA}} \\
\text{ECM}^* &= \text{ECM} \\
\text{DNAdam}^* &= \text{DNAdam}
\end{aligned}
$$

The system comprises six species NICD, Notch, TP53, TP63_TP73, miRNA, and EMTreg. ECM and DNAdam are input parameters that do not change during the simulation. For example, the model has been used in [55] to investigate the effect of Notch upregulation and TP53 deletion. The model captures the EMT dynamics triggered by TP53 deletion and Notch activation, and the interplay between multiple interactions that lead to mesenchymal behavior in epithelial mouse cells. Such mechanisms are also common in many other cancers.

Let us now assume, that this set of rules is the underlying truth for our mechanism inference algorithm, and therefore refer to the model as EMT-O (original). We have generated reference data by running the asynchronous updating simulator 100 times for each of the $2^8 = 256$ states and recorded the steady state for each run. We use this data, to automatically infer a set of rules.

Table 4 summarizes the so obtained results.

Since ECM and DNAdam are parameters and can not change during the simulation, the network naturally divides into four mutually exclusive sub-networks depending on the parameter inputs

$$(\text{ECM}, \text{DNAdam}) \in \{(0,0), (0,1), (1,0), (1,1)\}.$$

**Table 4. Summary of the asynchronous updating results for the EMT model from [55] after 100 asynchronous updating simulations for each of the 256 possible initial states.** The order of species is (NICD, Notch, TP53, TP63_TP73, miRNA, EMTreg, ECM, DNAdam). The network splits into four mutually exclusive sub-networks depending on the two parameters ECM and DNAdam. Initial states for three of those sub-networks can run into two different attractors with varying frequency.

| (ECM, DNAdam) | SS | frequency |
|---|---|---|
| (0, 0) | (0, 0, 0, 0, 0, 0, 0, 0) | 0.61 |
| | (0, 0, 1, 0, 1, 0, 0, 0) | 0.39 |
| (0, 1) | (0, 0, 1, 0, 1, 0, 0, 1) | 1.0 |
| (1, 0) | (0, 0, 1, 0, 1, 0, 1, 0) | 0.52 |
| | (1, 1, 0, 0, 0, 1, 1, 0) | 0.48 |
| (1, 1) | (0, 0, 1, 0, 1, 0, 1, 1) | 0.79 |
| | (1, 1, 0, 0, 0, 1, 1, 1) | 0.21 |

Each of those sub-networks contains the 64 states with the corresponding parameters fixed in each. In three of the four networks, the data suggests that initial states can run into two different steady states. A closer inspection reveals, that the system has in total three different attractors: (NICD, Notch, TP53, TP63_TP73, miRNA, EMTreg) = (0, 0, 0, 0, 0, 0), (0, 0, 1, 0, 1, 0), or (1, 1, 0, 0, 0, 1). Note, however, that for our method we treat the parameters as species and therefore work with seven different attractors. A schematic depiction of the resulting four sub networks divided by the corresponding parameter set can be found in S2 File. The input file for our method consists of a list for every one of the 256 states and their corresponding attractors.

We can now run the Boolean rules generator described in the previous section. Due to the many species involved in this system, we do not expect our method to produce rules that are short and easily understandable (at least not without additional constraints). However, they are created automatically and we could immediately put the generated rule-set into the asynchronous simulator and analyze the inferred model.

In the supplemental S3–S6 Figs, we depict the four generated subnetworks determined by the four different parameter options generated by our method. The blue lines are transitions that are captured in the original model (EMT-O) [55] as well as our forward path model (EMT-FW). The orange lines are transitions that are captured by EMT-FW but not EMT-O. The green lines are transitions that are not captured by EMT-FW, but are added by the backward model (EMT-BW). We want to point out, that EMT-O introduced some states that are part of a dual-attractor network, but only reach one of the steady states (denoted with the bright blue lines). Our implementation recognizes those states and successfully treats them in the same way.

In the supplementary Fig 4S, we can observe the difference between EMT-FW and EMT-BW for a network with a single steady state more clearly. Due to the large number of species, the inter-connectivity between the states results in backward transitions even in EMT-FW (thus a forward connection for one rule can act as the backward connection for another). The difference between the two networks only occurs between level 1 and level 2 of the distance to the steady state. Note, that for some states, EMT-O includes these transitions that are not captured by EMT-FW. Since the rulesets are very long, we put them into the supplement S1 File. As we can see, the terms are now too complicated to actually get a meaningful chemical kinetics interpretation.

**Comparison between data and automatically created networks.** Since the construction of EMT-BW includes all possible transitions under the asynchronous updating scheme, we need to remove particular transitions to recapture EMT-O. In the supplementary S2 File, we

**Table 5. Summary of transitions assigned to each network.** Each species in each sub-network has a number of transitions. The number on the left is the number of transitions for EMT-O, the number on the right is the number of transitions for the backward pathway model generated by our tool. The last row denotes the percentage for all four sub-networks in total.

| (ECM, DNAdam) | NICD | Notch | TP53 | TP63_TP73 | miRNA | EMTreg |
|---|---|---|---|---|---|---|
| (0, 0) | 32\|61 | 32\|61 | 32\|56 | 32\|56 | 28\|56 | 24\|57 |
| (0, 1) | 32\|63 | 32\|63 | 32\|63 | 28\|63 | 36\|63 | 24\|63 |
| (1, 0) | 32\|62 | 32\|62 | 32\|59 | 32\|59 | 28\|59 | 24\|59 |
| (1, 1) | 32\|60 | 32\|62 | 32\|57 | 28\|57 | 32\|60 | 24\|57 |
| | 52.03% | 51.61% | 54.47% | 51.06% | 52.1% | 40.68% |

list the full list of transitions to be removed from EMT-BW to capture the original EMT model, and depict the resulting network graph separated by the corresponding parameter set. In Table 5, we summarize the transitions by counting how many of them to remove from EMT-BW to obtain the original model.

The numbers on the left denote the number of transitions for EMT-O, while the numbers on the right are the corresponding numbers of transition we obtain with the backward rule generator. These numbers confirm our suspicion, that our generated rules include about twice as many transitions as EMT-O.

Let us now look at some simulation results. In Fig 4, we depict the dynamics of two initial states (NICD, Notch, TP53, TP63_TP73, miRNA, EMTreg, ECM, DNAdam) = (0, 0, 0, 0, 0, 0, 1, 0), and (0, 1, 0, 0, 0, 1, 0, 0) for the original model (row 1), and EMT-BW (row 2). For the first initial condition, EMT-O reaches the steady state (1, 1, 0, 0, 0, 1, 1, 0) more than half the time and the steady state (0, 0, 1, 0, 1, 0, 1, 0) less than half the time. For EMT-BW this dynamics is exactly reversed. For the second initial state reaching the attractor (0, 0, 1, 0, 1, 0, 0, 0) is significantly less likely in EMT-O than in EMT-BW, but qualitatively the correct behavior is obtained.

To get a broader view, we extended Table 4 by the corresponding frequencies for our automatically created systems in Table 6. We interpret that the subnetwork (ECM, DNAdam) = (1, 0) experiences a trend towards steady state (1, 1, 0, 0, 0, 1, 1, 0) for the systematic approach, where for EMT-O, both steady states of the subnetwork are reached about the same number of times. For the other three subnetworks, the frequencies to enter into a given steady state generally match well with EMT-O.

**Automated model selection.** Our automatic rule-creation method does not take the time evolution of the system into account. It is therefore not surprising, that our results experience some qualitative differences between the input model and our created models.

If, however, a model predicts the majority of a cell fate as death, when the experiment clearly states the majority as survival, the model is very restricted in its usefulness. In this section, we therefore propose an automatic model selection algorithm to find a model that agrees better with the underlying data.

If we start with the backward pathway model, we already cover all possible connections in the system. From Table 5, we know that we would need to remove about half of the connections from each network. This, however, is information taken from the underlying truth and can not be assumed as knowledge in a real experiment.

Our hypothesis is, that by removing a transition from the pathway towards an attractor, the initial state has to take a "detour" through the network to reach the attractor, thus making it less likely to reach this particular steady state, and more likely to move towards the other one instead. Due to the highly interconnected structures of our networks, however, we are aware that by removing the transition for one state to make it harder to reach an attractor, we might

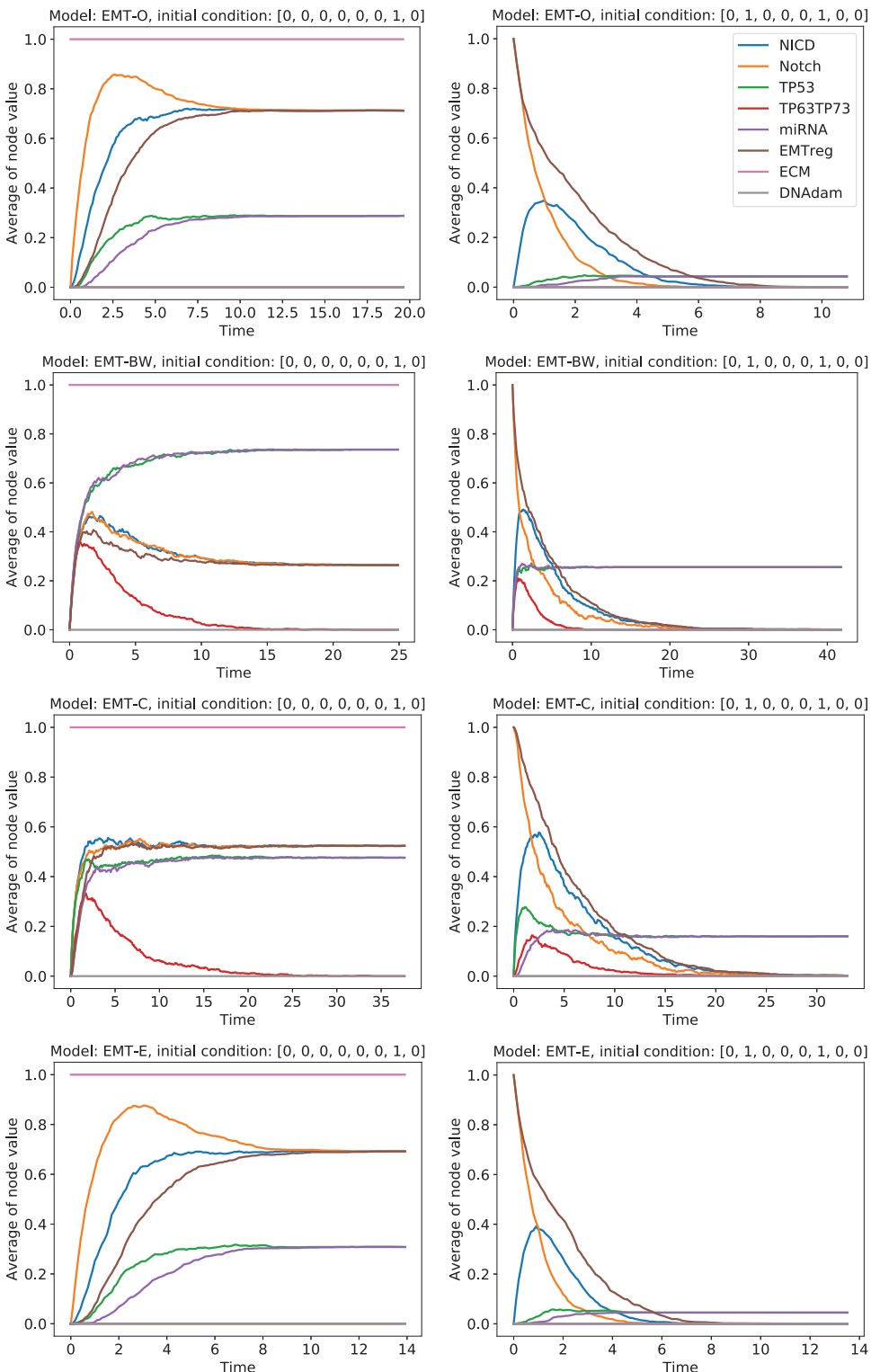

**Fig 4. Comparison between EMT-O and some of our automatically created rulesets (EMT-BW, EMT-C, EMT-E).** Initial state (0, 0, 0, 0, 0, 0, 1, 0) has the two attractors (0, 0, 1, 0, 1, 0, 0, 0) and (1, 1, 0, 0, 0, 1, 1, 0) (left column). The initial state (0, 1, 0, 0, 0, 1, 0, 0) converges towards the two attractors (0, 0, 1, 0, 1, 0, 0, 0) and (0, 0, 0, 0, 0, 0, 0, 0) (right column). On top, we depict the dynamics of the original model EMT-O. From row 2 to row 3 and 4, we include the more and more sophisticated automatically created network (EMT-BW, EMT-C, and EMT-E, respectively).

**Table 6. Summary of the asynchronous updating results after 100 asynchronous updating simulations for each of the 256 possible initial states.** The numbers on the right hand side are frequencies that the simulation reaches the corresponding steady state for the EMT model from [55], EMT-FW, and EMT-BW, respectively. The order of species is (NICD, Notch, TP53, TP63_TP73, miRNA, EMTreg, ECM, DNAdam).

| (ECM, DNAdam) | SS | EMT-O | EMT-FW | EMT-BW |
|---|---|---|---|---|
| (0, 0) | (0, 0, 0, 0, 0, 0, 0, 0) | 0.61 | 0.51 | 0.52 |
| | (0, 0, 1, 0, 1, 0, 0, 0) | 0.39 | 0.49 | 0.48 |
| (0, 1) | (0, 0, 1, 0, 1, 0, 0, 1) | 1.0 | 1.0 | 1.0 |
| (1, 0) | (0, 0, 1, 0, 1, 0, 1, 0) | 0.52 | 0.73 | 0.72 |
| | (1, 1, 0, 0, 0, 1, 1, 0) | 0.48 | 0.27 | 0.28 |
| (1, 1) | (0, 0, 1, 0, 1, 0, 1, 1) | 0.79 | 0.78 | 0.77 |
| | (1, 1, 0, 0, 0, 1, 1, 1) | 0.21 | 0.22 | 0.23 |

involuntarily also affect states that are supposed to reach the attractor more often. Therefore, a gradient based optimization will not perform well.

We chose a genetic optimization algorithm instead and extended our implementation by an option to randomly remove a number of transitions from the transition lists of each species. After gathering all the transitions created by EMT-BW and before we translate the transition lists into the ruleset, we randomly remove entries from those lists. The number of transitions to remove is the parameter that our optimizer chooses in order to improve the frequencies of the steady states.

This setup leads to a couple of difficulties.

1. By removing transitions, we could violate point 3 of our problem list above and remove the pathway necessary to reach the attractor. Before we remove any transition from a species list, we therefore first check, if removing this transition is legal. Only if the reachability check answers *True*, the transition will be removed from the list. We therefore ensure that the initial states will always be able to reach their attractors. These checks, however, extend the runtime for the creation of each model. Furthermore, we do not always remove the number of transitions proposed by the optimizer. If, for example, the optimizer proposes to remove 50 transitions and the algorithm can only find 40 valid transitions to remove, it will stop at the removal of 40 transitions, while still registered as a model with 50 transitions removed.

2. Due to the randomness of the removal, it is not enough to only create one model according to the suggested number of transition removals. In a genetic optimizer, every generation consists of multiple individuals that suggest their own number of transitions to remove. One of the individuals might have found the perfect number, however, the randomly selected transitions might be a bad choice and thus result in a bad fit. To avoid this, each individual of the algorithm does not only simulate one model, but multiple models with the same number of transitions removed.

A genetic algorithm consists of a number of individuals called a generation. After each individual computes its fitness, a selection process decides how to pick individuals to mate with each other and produce two new individuals according to the selected crossover. Some parts of the new individuals also get mutated according to the given mutation percentage.

For our optimization, our algorithm is based on the software package DEAP [57] using the build-in toolboxes for crossover, mutation, and selection.

We initialize 150 individuals as lists of random numbers between 0 and 1, denoting the number of transitions to be removed. Then, each individual uses its list of numbers to create

50 models that differ from each other by the randomness of the transition removals. Each of these models then runs 100 asynchronous updating simulations for each of the 256 initial states. It gathers the resulting frequencies of each state and compares it with the corresponding frequencies of the experimental data (i.e., the data taken from the model from [55]). The root mean square error (RMS) is computed by taking the difference between each of the corresponding frequencies. The fitness for each individual is the smallest RMS from the 50 simulations.

The detailed parameter setup of DEAP for our simulation can be found in the supplementary file S1 Table. In the supplementary S7 Fig, we see the development of the RMS over 90 generations. This simulation creates a total of $150 \times 50 \times 90 = 675000$ models. We identify the smallest RMS at the 86th generation with a value of 12.68. This will be from now on referred to as model EMT-C. A similar RMS of 12.76 can be found at generation 40 (EMT-B), which is less than half of the full simulation. As a third point of interest, we chose a relatively early model found at generation 7 with an RMS of 13.14 (EMT-A).

**Model selection using expert knowledge.** The above introduced model selection is the most general version with the least amount of knowledge input possible. In this section, we decrease the state space of models by adding expert knowledge to the interference process. Often the network structure (i.e. the dependence of a rule on other species) is known, or at least suspected. In our situation, the rule for NICD, e.g., displays a dependency to Notch, TP63_TP73, and TP53, but not to miRNA, EMTreg, ECM, or DNAdam. We now assume here that we know the dependencies for each node. In other words, the expression of our new rule set is no longer allowed to include the dependency of a species that it does not depend on in the original rule set. This is an assumption that we can impose upon a Boolean ground truth. Note, that in the case of our enzyme-substrate example, we are reconstructing a kinetic model rather than a Boolean model. We therefore can not directly use this approach, since the dependencies in this case are unknown.

For the EMT model, however, we can take a closer look into how the ruleset is formulated from the transition states that have been determined by our model creator. Similar to the automated model selection from the previous section, we select our new network by legally discarding transitions. In this case, however, we do not randomly choose a set, but exploit the fact that $(x \wedge y \wedge z) \vee (\overline{x} \wedge y \wedge z) = y \wedge z$. I.e., if we want to eliminate a dependency of a variable, we need to eliminate each transition that does not include its symmetric counter part in the transition list. This method assures a non-dependency of the right hand side on the chosen species.

Note, however, that our rules are formulated by species$^* = \left( \underset{\text{transitions}}{\text{V}} \text{ transition state} \right) \underline{\vee}$ species.

Due to the *XOR* operation, we need to treat the self-dependency in the opposite manner. While for the elimination process of the other species we want to make sure that the symmetric counter parts are all present in the transition list, to ensure non self dependency we want to make sure that only one state of the two counter parts is present. In this step we need to be careful to ensure the symmetry from the step before. Let us, e.g,. assume that we want to create the first rule for species $A$ for a list of five species $(A, X, Y, Z, W)$, that only depends on the second and third species $X$ and $Y$. Let us further assume, that the last two species, $Z$ and $W$, have been eliminated from the formulation by only keeping the symmetric counterparts of each transition state in the list. In addition, both states $(0, X, Y, Z, W)$ and $(1, X, Y, Z, W)$ are in the transition list. We therefore need to eliminate one of those states. The algorithm needs to choose, whether to eliminate the transition, where the first species is in state 0, or in state 1. Both choices lead to a correct network. However, to guarantee the symmetry, for a fixed pattern of $X$ and $Y$, the same choice has to be made. Eliminating transition states this way, however, might result in the loss of the connection to a steady state. We therefore include the reachability check that we

already discussed in the automated model selection in this type of optimization as well. If reachability is not satisfied for any of the possible networks, the assumptions made by the user are not compatible with the biological observations of the experiment. In this case, relaxed assumptions have to be applied. The selection process is detailed in Algorithm 1.

**Algorithm 1:** Model selection using biological insight.

```
start with all possible transitions EMT-BW;
while select species to make rule for do
  while select species that must not be part of the rule, except self
reference do
    If symmetric counterpart not in transition list then
      eliminate transition from list;
    else
      keep both transitions in list;
    end
  end
  if both symmetric counterparts regarding the selected species are in
the transition list then
    choose a pattern for the non-eliminated species;
    decide the state of the species to remove (either 0 or 1);
    for every transition state of the same pattern, eliminate the tran-
sition where the species is in the chosen state;
  else
    keep the transitions;
  end
end
```

Due to the non-unique choice of the asymmetric elimination process taking place for the self-elimination, this algorithm leads once again to multiple possibilities of the network structure. In Table 7, we give an overview of the species to eliminate and the resulting number of possibilities for each rule.

From there we can see that the rules for Notch and EMTreg are both unique. There are, however, still >8e6 network possibilities for the rule for TP53 alone. Since each of these transition lists can be combined with every other transition list, this method still leads to a total of $16 \times 1 \times 8388608 \times 256 \times 32 \times 1 > 1e12$ possible networks to choose from. We therefore used again the DEAP algorithm to optimize for the frequencies of the steady states. For the list of the used DEAP parameters see the supplementary file S1 Table. In the supplementary S8 Fig, we show the results for running the optimization over 25 generations using a population of 150 individuals. We already reach an RMS of 7.45 at generation 14, however, the smallest RMS achieved in this optimization is 7.44 at generation 20. This is the model that we use as expert guided model (EMT-E) in the following analysis. Note, that for this optimization, there was no

**Table 7. Overview of model selection for the expert knowledge guided variant.** The numbers are obtained using Algorithm 1, starting with the transition list of the full backwards model and removing every possible combination of the elimination of self reference by brute force.

| rule | species to eliminate | possible number of resulting networks |
|---|---|---|
| NICD | NICD, miRNA, EMTreg, ECM, DNAdam | 16 |
| Notch | NICD, Notch, TP53, TP63_TP73, EMTreg, DNAdam | 1 |
| TP53 | Notch, TP53, ECM | 8388608 |
| TP63_TP73 | Notch, TP63_TP73, EMTreg, ECM | 256 |
| miRNA | NICD, Notch miRNA, ECM, DNAdam | 32 |
| EMTreg | Notch, TP53, TP63_TP73, EMTreg, ECM DNAdam | 1 |

additional random choice necessary. This simulation therefore produced $150 \times 25 = 3750$ models, where some of them are equivalent.

The resulting rule set for this choice is

$$
\begin{aligned}
\text{NICD}^* \; = \; & (\text{Notch} \wedge \overline{\text{TP53}} \wedge \overline{\text{TP63\_TP73}}) \\
& \vee (\overline{\text{Notch}} \wedge \text{TP53} \wedge \text{TP63\_TP73}) \\
\text{Notch}^* \; = \; & \text{ECM} \wedge \overline{\text{miRNA}} \\
\text{TP53}^* \; = \; & (\text{DNAdam} \wedge \text{EMTreg} \wedge \overline{\text{miRNA}} \wedge \text{NICD} \wedge \text{TP63\_TP73}) \\
& \vee (\text{DNAdam} \wedge \overline{\text{EMTreg}} \wedge \text{miRNA}) \\
& \vee (\text{DNAdam} \wedge \overline{\text{EMTreg}} \wedge \overline{\text{NICD}}) \\
& \vee (\text{DNAdam} \wedge \text{miRNA} \wedge \overline{\text{NICD}}) \\
& \vee (\text{DNAdam} \wedge \overline{\text{NICD}} \wedge \overline{\text{TP63\_TP73}}) \\
& \vee (\overline{\text{DNAdam}} \wedge \overline{\text{EMTreg}} \wedge \overline{\text{miRNA}} \wedge \text{NICD}) \\
& \vee (\overline{\text{EMTreg}} \wedge \text{miRNA} \wedge \overline{\text{TP63\_TP73}}) \\
& \vee (\overline{\text{EMTreg}} \wedge \overline{\text{miRNA}} \wedge \overline{\text{NICD}} \wedge \text{TP63\_TP73}) \\
& \vee (\overline{\text{EMTreg}} \wedge \text{NICD} \wedge \overline{\text{TP63\_TP73}}) \\
\text{TP63\_TP73}^* \; = \; & \text{False} \\
\text{miRNA}^* \; = \; & \overline{\text{EMTreg}} \wedge \text{TP53} \\
\text{EMTreg}^* \; = \; & \text{NICD} \wedge \overline{\text{miRNA}} \\
\text{ECM}^* \; = \; & \text{ECM} \\
\text{DNAdam}^* \; = \; & \text{DNAdam}
\end{aligned}
$$

As we can see, the unique possibilities of the model selection for Notch and miRNA result in the correct rule expression. Setting TP63_TP73 to False is correct from a mathematical point of view (all attractors have this species at 0). This is a choice out of 256 possibilities and is valid given the system constraints imposed on the model selection. This rule could be made more biologically relevant by imposing additional expert knowledge, if desired. Due to the $>8e6$ possible formulations for TP53, the convoluted formulation is not very surprising. The randomly chosen models from the model optimizer above, as well as the original EMT-FW and EMT-BW models have even longer and more convoluted terms. The rules for NICD and miRNA are very close to the ones from the original paper. With the reduction of terms, the interpretation of the rules becomes easier as well.

**Analysis of the different models.** In Fig 4, we can observe how the optimization gradually improves the model. While EMT-BW produces some qualitatively wrong dynamics, EMT-C converges towards the correct dynamics. EMT-E captures the dynamics nearly perfectly. For a full comparison of the dynamics of all initial values, we refer the reader to the additional S3–S6 Files. To get an overall idea of how the dynamics of the states develop, we extend Table 6 by the frequencies resulting from the three models with EMT-A, EMT-B, and EMT-C, as well as the result for the expert knowledge guided optimization EMT-E, respectively in Table 8. As we have observed before, the steady state for the parameter set (ECM, DNAdam) = (0, 0) for the EMT model is biased towards the first state (NICD, Notch, TP53, TP63_TP73, miRNA, EMTreg, ECM, DNAdam) = (0, 0, 0, 0, 0, 0, 0, 0) with approximately 60%. Both the EMT-FW and EMT-BW, as well as our first selected model EMT-A are very close in their behavior to

**Table 8. Summary of the asynchronous updating results after 90 asynchronous updating simulations for each of the 256 possible initial states.** The numbers on the right hand side are frequencies that the simulation reaches the corresponding steady state for the EMT model from [55]. This is an extension to Table 6 with the inclusion of the three selected models from the genetic optimizer EMT-A, EMT-B, and EMT-C, respectively. Furthermore, we include the model found by the expert knowledge guided optimization EMT-E.

| (ECM, DNAdam) | SS | EMT-O | EMT-FW | EMT-BW | EMT-A | EMT-B | EMT-C | EMT-E |
|---|---|---|---|---|---|---|---|---|
| (0, 0) | (0, 0, 0, 0, 0, 0, 0, 0) | 0.61 | 0.51 | 0.52 | 0.59 | 0.66 | 0.6 | 0.63 |
| | (0, 0, 1, 0, 1, 0, 0, 0) | 0.39 | 0.49 | 0.48 | 0.42 | 0.34 | 0.4 | 0.37 |
| (0, 1) | (0, 0, 1, 0, 1, 0, 0, 1) | 1.0 | 1.0 | 1.0 | 1.0 | 1.0 | 1.0 | 1.0 |
| (1, 0) | (0, 0, 1, 0, 1, 0, 1, 0) | 0.52 | 0.73 | 0.72 | 0.5 | 0.5 | 0.5 | 0.5 |
| | (1, 1, 0, 0, 0, 1, 1, 0) | 0.48 | 0.27 | 0.28 | 0.5 | 0.5 | 0.5 | 0.5 |
| (1, 1) | (0, 0, 1, 0, 1, 0, 1, 1) | 0.79 | 0.78 | 0.77 | 0.75 | 0.76 | 0.77 | 0.8 |
| | (1, 1, 0, 0, 0, 1, 1, 1) | 0.21 | 0.22 | 0.23 | 0.25 | 0.24 | 0.23 | 0.2 |

50%. With an RMS <13 (EMT-B, EMT-C, EMT-E), the models, however, capture the bias towards the first steady state with 60% very similar to EMT-O. Contrary to the predictions of EMT-FW and EMT-BW, that bias the steady states of the parameter set (ECM, DNAdam) = (1, 0) towards the steady state (NICD, Notch, TP53, TP63_TP73, miRNA, EMTreg, ECM, DNAdam) = (0, 0, 1, 0, 1, 0, 1, 0) with 70%, all of our selected model keep the ratio of approximately 50% towards both steady states similar to the EMT results.

The bias observed for the parameter set (ECM, DNAdam) = (1, 1) towards the steady state (NICD, Notch, TP53, TP63_TP73, miRNA, EMTreg, ECM, DNAdam) = (0, 0, 1, 0, 1, 0, 1, 1) is observed by all the models.

Looking at the dynamics for parameter set (ECM, DNAdam) = (1, 0), we immediately see the effect of the optimizer finding the trend of no bias towards a steady state compared with the non-optimized models. In Fig 5, we look into the distributions of the models according to Table 8. We only depict one of the steady states, since the other state would only be a complement to the corresponding figure.

The Kolmogorov-Smirnov test tries to evaluate whether two data sets are drawn from the same distribution. A small p-value therefore hints towards two different distributions underlying the data sets. In Fig 5, we see for the subnetwork (ECM, DNAdam) = (0, 0), that the distributions of EMT-FW and EMT-BW hint towards a different underlying distribution than the data drawn from EMT-O. The generally selected models with EMT-A, EMT-B, and EMT-C have a relatively large p-value and therefore hint towards a similar distribution to the data for EMT-O. For the expert knowledge guided EMT-C model, the p-value is close to 1, showing that we have obtained excellent agreement.

A similar behavior can be observed for the subnetwork (ECM, DNAdam) = (1, 0). However, the p-values for EMT-FW and EMT-BW are even smaller, while for the optimized models, the $p$ value is relatively large. This means that performing the optimization is more important for this subnetwork.

For the subnetwork (ECM, DNAdam) = (1, 1) we see that EMT-FW and EMT-BW give a relatively large p-value compared to the other two subnetworks—they can still be considered small, but are far from significant. These models, therefore, might already give a decent approximation to the underlying data set. The optimization, in this case, actually gives a worse result for the models EMT-A and EMT-B. The effect of the optimizations for EMT-C and EMT-E are not as strong as for the other subnetworks, but both optimization results can still be seen as an improvement to EMT-FW and EMT-BW.

The dynamics for all models discussed in this chapter for steady state 1, 2, 3, and 4 can be found in S3–S6 Files, respectively. Note, that for steady state (ECM, DNAdam) = (0, 1)

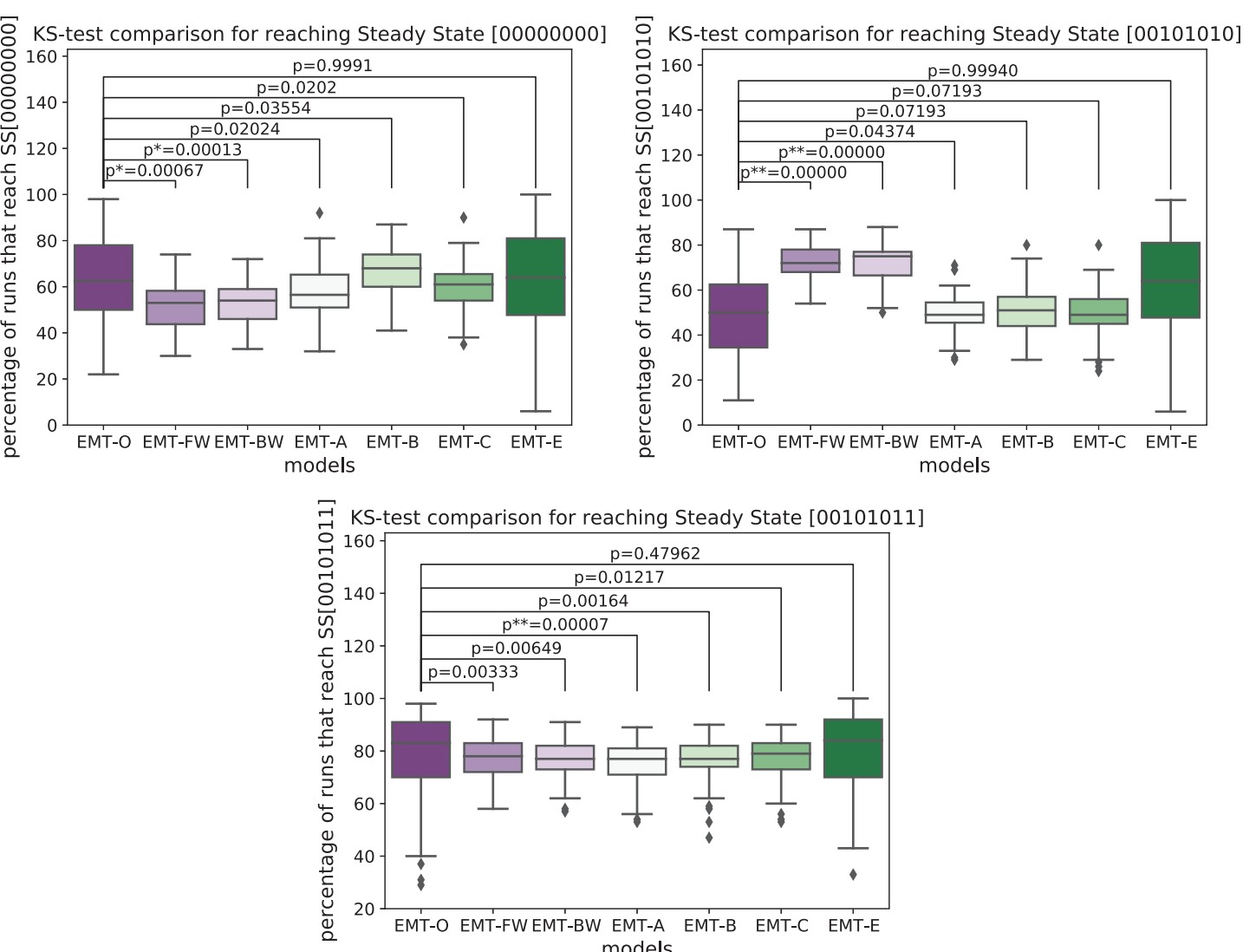

**Fig 5. Frequency distribution of the states divided by the corresponding subnetworks.** The second steady state for each of those subnetworks is omitted, since it is the complement of the depicted plots here. The noted p-value is taken from a Kolmogorov-Smirnov test between the distribution of frequencies between the original model EMT-O and EMT-FW, EMT-BW, EMT-A, EMT-B, EMT-C, and EMT-E, respectively. The top left figure depicts the distribution of all states that can reach the attractor (00000000). The figure in the top right depicts the distribution of all states that can reach the attractor (00101010). The figure on the bottom depicts the distribution of all states that can reach the attractor (00101011).

(S4 File), the initial states all converge towards a single attractor. Since the corresponding steady state is unique, EMT-BW, EMT-A, EMT-B, and EMT-C are equivalent (up to random fluctuations in the stochastic solver).

To summarize those findings, we look at Fig 6, where we explore in detail the differences between EMT-O and our automatically generated models. In the first category <10, we include all the states that are within 10% of the EMT model. For example, if a state in EMT-O reaches a steady state 70%, and our selected model reaches the same state 65% of the time, we compute the distance using $70 - 65 = 5$. A distance of 5 is smaller than 10 and thus can be considered as a good state that represents a similar behavior than EMT-O.

Category (10, 19) counts all the states that are within 20% of the EMT results, and also experience the same qualitative behavior. If, e.g., a state in EMT-O reaches a steady state 70%, and

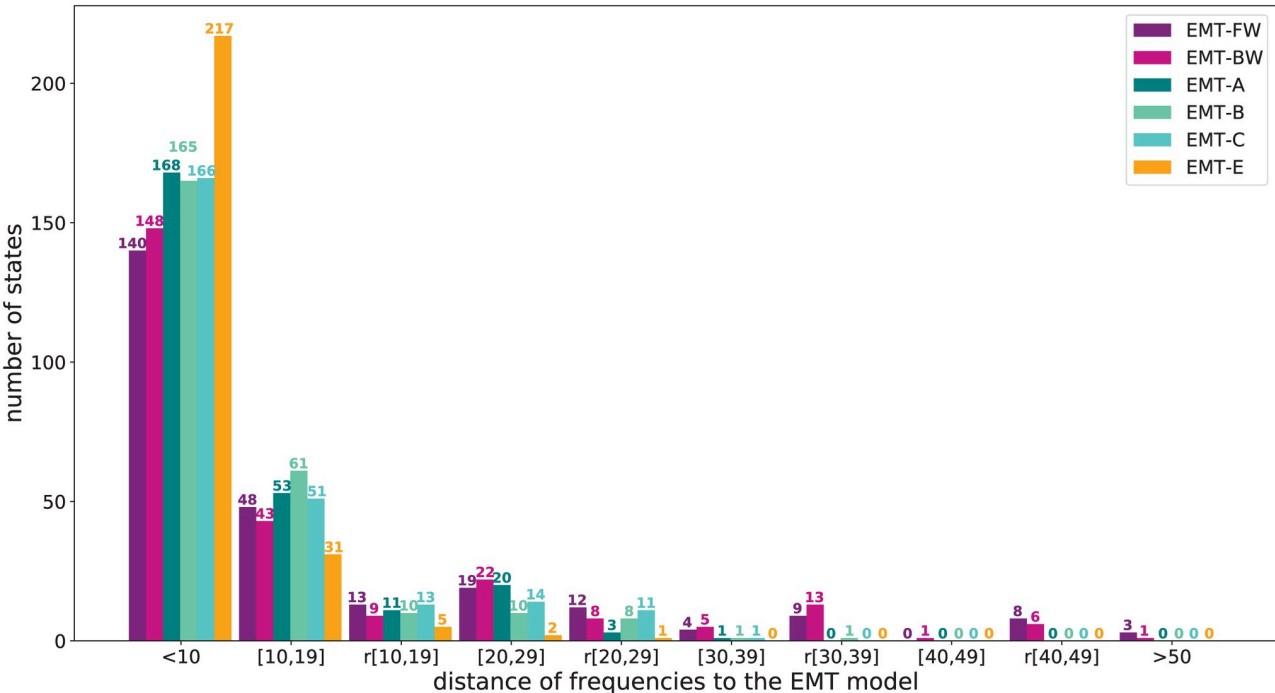

**Fig 6. Accuracy of the steady state frequencies in relation to the EMT-O.** The categories on the x-axis are in reference to the distance of the frequencies resulting from the EMT model over all states that converge to 2 steady states. In category <10, all states are counted that are within a distance of 10% of the frequencies resulting from EMT-O. Category (10, 19) counts the number of states that are within a distance of 20% from the results from EMT-O with the same qualitative behavior. In category r(10, 19), we count the number of states that are within a distance of 20% from the results from EMT-O that also have a reversed qualitative behavior. Categories (20, 29) and r(20, 29) count the number of states that are within 30% of the EMT states with the same qualitative behavior, and the reversed qualitative behavior respectively, similar to categories (30, 39), r(30, 39), (40, 49), and r(40, 49). In category >50, we count all the states that are more than 50% away from the EMT-O frequencies.

the corresponding model state reaches this state 55%, the distance of these states is 70 − 55 = 15. Both states have a bias towards this steady state (i.e., they are both larger than 50%), and therefore this state gets sorted into this category.

If the EMT-O state reaches a steady state 60%, and the corresponding model state reaches the same steady state 45%, the distance is still 60 − 45 = 15, the qualitative behavior, however, is now reversed (60 > 50, but 45 < 50), and this state gets sorted into category r(10, 19). On first glance, the qualitative behavior of the states seems like a more important metric to keep on the states behavior. Note, however, that for a state to be within 20% of the EMT states and have a reversed behavior, both frequencies have to be relatively close to 50% and thus can still be considered approximately half.

The larger the distance, the more significant the impact of qualitatively wrong behavior gets as well. Therefore, the higher the bar on the left, i.e, the more states that are close to the EMT frequencies, the better the model. In this figure, we include now all the models discussed in this section. The purple colors depict the non-optimized models EMT-FW and EMT-BW. We can see, that both models experiences a couple of states, that are more than 50% away from the behavior of the EMT model. Neither of our optimized models has states in this category.

Our expert knowledge guided model has one state that is within 30% discrepancy that has the wrong qualitative behavior. 77% of states are within 10% of the original frequency values. This graph shows clearly, that most states are on the left side of the graph, i.e., close to the results of EMT-O, and that the optimization algorithm clearly skews the bars further to the

left. The 217 orange states in the category <10 demonstrate the power of additional knowledge to guide the model selection process.

## Discussion

We have proposed an algorithm to infer Boolean rules from the mapping of initial states to attractors. We have exemplified this method for two biologically relevant examples. Namely, a classic enzyme-substrate system and a model of Epithelial to Mesenchymal Transition (EMT) in cancer metastasis. In both cases the algorithm, without providing any additional information, provides candidate models that match the dynamics of the underlying system well. In particular, the steady states and their respective probability are reproduced accurately. For the enzyme-substrate system the dynamics is also well resolved. However, for the EMT model there is still an extremely large number of possible candidates and thus the dynamical behavior is not always faithfully resolved. This can be improved by incorporating additional insight into the systems. We have done this by constraining the network structure. That is, we have made assumptions on the species that each Boolean rule depends on. This results in a Boolean network which extremely accurately describes the dynamics of the underlying model. In fact, some of the inferred Boolean rules are identical to the ground truth.

The proposed algorithm could be used to infer the mechanisms of signaling, gene-regulatory, and any other input-output processes in an automated manner. This enables us to use our methodology as part of a larger data processing, model inference, and prediction framework that can be used without human intervention. In this work we have exclusively considered data that only model the initial and final state of the system, because such experimental data are commonly available [15]. However, with ever advancing measurement techniques, time series information is becoming increasingly more available for such systems. We envisage the use of such time series to further improve model selection, which will be the subject of future research.

One limitation of our method is the necessity to map every possible state towards at least one steady state. Knowledge from experimental data may not provide this information, thus necessitating modeling assumptions. To address these limitations, users could first map all known states to their corresponding known steady states. Next, a reachability check is necessary to determine whether these states have a possible pathway available to reach their steady states. If this is not the case, other states may have to be assigned to build a pathway. However, the user must ensure that these states do not enable pathways towards undesired steady states. Other states that have not been assigned at this point could be incorporated into the network in multiple ways. The least biased method would involve assigning them to as many steady states as possible, as long as this does not enable unwanted connections towards an undesired attractor. The choice of the attractor and state association can also be made by the proximity to the given attractors in the system. Yet another option could be made to exclude a given state from the network by not assigning it to any attractors at all. Since all these choices are model assumptions, they could also be incorporated by the model selection process as an additional parameter.

In this work, we used the General Asynchronous updating scheme as, e.g., proposed in [23, 50–53] to simulate network dynamics. As discussed in the Enzyme-Substrate dynamics section, this corresponds with the model assumption that all reactions are equally likely to occur, which is not always a reasonable approximation for biological processes. Different biological processes, such as signal transduction events and transcriptional events, occur within different time frames. To account for different time scales, the asynchronous updating scheme can be ranked according to the biological time frame, as suggested in e.g., in [58]. Another way of

adding more biological relevance to the updating scheme is to weight activation actions and inhibition actions of a node differently as proposed, in e.g. [59]. With this strategy, the activation and degradation propensities are taken into account in a manner similar to their formulation in the chemical master equation. Using a differently weighted asynchronous updating scheme can have a significant effect on the simulated frequencies. We note, however, that in our setup, the asynchronous update simulator is an exchangeable simulator and that the optimization algorithm is still perfectly usable with a different simulation setup.

Finally, our work takes advantage of parallel computing environments, thus reducing the amount of time required to enumerate logic rules by hand. We believe that computer-driven mechanism exploration coupled with a model selection, such as that presented in this work, could be a highly suitable tool to advance mechanism exploration and accelerate hypothesis prediction and testing *in silico* for experimental validation, thus reducing the time and effort required to obtain mechanistic knowledge from experimental data.

## Conclusion

We presented a general-purpose algorithm for mechanism exploration, hypothesis exploration, and model selection using initial and attractor state data and high-performance computing. We demonstrated the mechanism of our method using the simpler example of enzyme-substrate kinetics, and extended our method to the larger and biologically relevant EMT model. We have shown how to improve the obtained models by including an unbiased genetic optimization pipeline for a model selection process. To decrease the resulting search space for this process, we also suggested a way to incorporate expert biological knowledge as part of the optimization process, which was able to obtain an appropriate model in a significantly shorter timespan. We therefore conclude that our approach greatly accelerates the inference of logic-based rules for complex biochemical networks and leads to dynamic networks that can be further explored in order to obtain testable hypotheses.

## Extended methods

The generic model selection consists of a genetic algorithm population of 150 individuals. Each of the individuals performs 50 simulations to account for the randomness of transition removal. We sequentially initialize the optimization with a Python interface and spawn a parallel environment using 50 nodes on an IBM power 9 architecture. On each node, one of the 50 individuals create and simulate the model according to the random process of transition removal. These simulations are completely independent and thus scaling is only restricted by the size of the population of the genetic algorithm. To speed up the simulations, each of the created rule sets was compiled into a C++ code to perform the asynchronous updating simulations. We measured a computational wallclock time for approximately 6 hours for one generation of the algorithm. The full code can be accessed at https://github.com/LoLab-VU/Boolean_rules_creator.

## Supporting information

**S1 File. Demonstration of Algorithm 1, and Rulesets for EMT-FW, EMT-BW, EMT-A, EMT-B, and EMT-C.**
(PDF)

**S2 File. Lists all states that have to be taken from the full backward model to obtain the original model from the paper.** The states are in the form of lists, as well as the graphic

representation of the network.
(PDF)

**S3 File. Displays all the simulation results for the EMT network that reach steady state 1: (ECM, DNAdam) = (0, 0).**
(PDF)

**S4 File. Displays all the simulation results for the EMT network that reach steady state 2: (ECM, DNAdam) = (0, 1).**
(PDF)

**S5 File. Displays all the simulation results for the EMT network that reach steady state 3: (ECM, DNAdam) = (1, 0).**
(PDF)

**S6 File. Displays all the simulation results for the EMT network that reach steady state 4: (ECM, DNAdam) = (1, 1).**
(PDF)

**S1 Fig. Graphic representation of how to achieve the transition table for the enzyme-substrate dynamics for states that have a Hamming distance of 2 away from the steady state (E, S, ES, P) = (1, 0, 0, 1) marked in red.** The states of Hamming distance 2 to the attractor are marked in pink. Each of these pink states has the possibility to switch one of the species E, S, ES, and P. If the bit flip results in a possible pathway, the new state is colored in green, otherwise it stays black. For the state (E, S, ES, P) = (0, 0, 1, 1), e.g., the flip of bit E results into the state (1, 0, 1, 1), which is a Hamming distance away from the attractor, and therefore a viable pathway. The state (1, 0, 1, 1) is therefore marked in green. Since state (0, 0, 1, 1) needs to swap the E-bit, this state will get added to the transition list for species E (see Table 3 of the main paper). For this pathway to reach the destination, the state (1, 0, 1, 1) needs to flip its third bit (denoted in green over the arrow) to reach the red attractor (therefore, (1, 0, 1, 1) is sorted in the transition list for ES in Table 3). The flip of S and P result in the states (0, 1, 1, 1) and (0, 0, 1, 0) respectively, which are a Hamming distance of 3 away from the attractor. This would mean that the pathway is going away from the attractor, and therefore these states are not considered in the forward only algorithm. Therefore, these states are denoted in black. Note, however, that these states would be sorted into the backward pathways algorithm. Flipping the bit for ES results in the state (0, 0, 0, 1). This state has been eliminated as initial condition since a system in which only P exists is not meaningful. This path therefore will also not be sorted in any of the transition lists and therefore stays black.
(PDF)

**S2 Fig. This figure is a continuation for the pathway construction started in S1 Fig.** On top, we see the possible evolution of states with a Hamming distance of 3 away from the attractor in blue. The state (E, S, ES, P) = (0, 0, 1, 0), e.g., can flip either the first or the fourth bit, to reach one of the pink states introduced in S1 Fig. The state (0, 0, 1, 0) gets therefore sorted into the transition list of E as well as P. It would be sorted into the S list, if we consider backward pathways as well, but flipping ES leads to an unfeasible state in the system, which is why this state is never sorted into the transition list for ES. On the bottom, we see the starting pathway for the state that is a Hamming distance of 4 away from the attractor and how to reach the Hamming distance states of 3 to create the desired pathway.
(PDF)

**S3 Fig. Network structure for the parameter set (ECM, DNAdam) = (0, 0) resulting from the rule set created by our tool.** The bright red circles are the target attractors of the system. The blue lines are the lines that occur in EMT-O, as well as in EMT-BW. The orange lines are transitions that are part of EMT-BW but not part of EMT-O. The green lines are transitions that are part of EMT-BW but not EMT-FW. The light blue lines depict states that are part of the network, but only have access to one of the attractors and not both of them.
(PDF)

**S4 Fig. Network structure for the parameter set (ECM, DNAdam) = (0, 1) resulting from the rule set created by our tool.** The bright red circle is the target attractors of the system. The blue lines are the lines that occur in EMT-O, as well as in EMT-BW. The orange lines are transitions that are part of EMT-BW but not part of EMT-O. The green lines are transitions that are part of EMT-BW but not EMT-FW.
(PDF)

**S5 Fig. Network structure for the parameter set (ECM, DNAFdam) = (1, 0) resulting from the rule set created by our tool.** The bright red circles are the target attractors of the system. The blue lines are the lines that occur in EMT-O, as well as in EMT-BW. The orange lines are transitions that are part of EMT-BW but not part of EMT-O. The green lines are transitions that are part of EMT-BW but not EMT-FW. The light blue lines depict states that are part of the network, but only have access to one of the attractors and not both of them.
(PDF)

**S6 Fig. Network structure for the parameter set (ECM, DNAdam) = (1, 1) resulting from the rule set created by our tool.** The bright red circles are the target attractors of the system. The blue lines are the lines that occur in EMT-O, as well as in EMT-BW. The orange lines are transitions that are part of EMT-BW but not part of EMT-O. The green lines are transitions that are part of EMT-BW but not EMT-FW. The light blue lines depict states that are part of the network, but only have access to one of the attractors and not both of them.
(PDF)

**S7 Fig. Span for the root mean square (RMS) error for each generation of the genetic optimization algorithm.** Each generation has 150 individuals and each individual chooses the minimum out of 50 independent runs for its RMS. Setup for the DEAP algorithms can be found in S1 Table. The orange line is largest RMS for each generation, and the blue line is the smallest RMS both chosen as the smallest number of 50 random simulations. The smallest RMS within 90 generations is 12.68 (EMT-C) at generation number 86. We reach a similar RMS at generation 40 with a value of 12.76 (EMT-B). An early fit after 7 generations already achieves an RMS of 13.14 (EMT-A).
(PDF)

**S8 Fig. Span for the root mean square (RMS) error for each generation of the genetic optimization algorithm for the human guided variant.** Each generation has 150 individuals. Setup for the DEAP algorithms can be found in S1 Table. The orange line is largest RMS for each generation, and the blue line is the smallest RMS. The smallest RMS within 25 generations is 7.44 at generation number 20 (EMT-E). The fit found before that is similarly high with a value of 7.45.
(PDF)

**S1 Table. Parameters for the DEAP algorithm.**
(PDF)

## Acknowledgments

We thank Prof. Vito Quaranta and the Quaranta Lab for their useful insights throughout the development of this work.

## Author Contributions

**Conceptualization:** Martina Prugger, Lukas Einkemmer, Samantha P. Beik, Perry T. Wasdin, Leonard A. Harris, Carlos F. Lopez.

**Funding acquisition:** Carlos F. Lopez.

**Software:** Martina Prugger, Lukas Einkemmer, Perry T. Wasdin.

**Supervision:** Carlos F. Lopez.

**Writing – original draft:** Martina Prugger, Lukas Einkemmer, Carlos F. Lopez.

**Writing – review & editing:** Martina Prugger, Lukas Einkemmer, Samantha P. Beik, Perry T. Wasdin, Leonard A. Harris, Carlos F. Lopez.

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
