## [Decision Letter · Decision Letter 0]

11 Feb 2021

Dear Dr. Lopez,

Thank you very much for submitting your manuscript "Unsupervised logic-based mechanism inference for network-driven biological processes" for consideration at PLOS Computational Biology.

As with all papers reviewed by the journal, your manuscript was reviewed by members of the editorial board and by several independent reviewers. In light of the reviews (below this email), we would like to invite the resubmission of a significantly-revised version that takes into account the reviewers' comments.

We cannot make any decision about publication until we have seen the revised manuscript and your response to the reviewers' comments. Your revised manuscript is also likely to be sent to reviewers for further evaluation.

Sincerely,

Jeffrey J. Saucerman

Associate Editor

PLOS Computational Biology

Jason Haugh

Deputy Editor

PLOS Computational Biology

Reviewer's Responses to Questions

**Comments to the Authors:**

Reviewer #1: This manuscript proposes an unsupervised method to infer Boolean (logical) functions from the knowledge of all initial states that can converge to an attractor. The work could be a useful contribution, complementary to other existing network inference approaches. Unfortunately, several aspects of the work are not clearly presented or not sufficiently addressed. In addition, the abstract contains statement that are not borne out in the manuscript.

Major comments:

1. The algorithm expects binary state information for every element (node) in the system. The two examples presented in the manuscript used all the 2^N states of the system. Some of these states were attractors and for all the remaining states the algorithm needed to know to which attractor they converge. There are at least two difficulties of generating this required input knowledge. How are the binary states decided from continuous experimental data? How realistic is to require that every state of the system was considered as an initial state in an experimental investigation? To my knowledge, it is generally assumed that there is a single, “natural” initial state. It is not general practice (and may be impractical) in experiments to start the system from every state it could have. How would the algorithm cope with having only a fraction of the information it needs?

2. The algorithm generates Boolean rules (functions) for each node. This is not the same as “mechanistic hypotheses of cellular processes” (as stated in the abstract). Consider the sets of Boolean functions inferred for the enzyme-substrate system (line 142, or 178, or 199). What are the mechanistic hypotheses expressed by these functions? Is the implicit assumption that a Boolean function is already a mechanistic hypothesis? If so, this should be stated explicitly and supported by evidence.

3. The method of generating the Boolean functions from the ordered state transitions is not explained clearly. In line 135, what is a “transition state”? what does it mean to “ translate every transition state via the logical operator AND” ? What does this mean: “An exclusive XOR with the active species ensures that the species activates only with the states from the given list.”?

4. The last sentence of the abstract states “We then conclude by placing our results in the context of ongoing work in the field and discuss how our approach could be used to infer mechanisms of signaling, gene-regulatory, and any other input-output processes describable by logic-based mechanisms. “ In fact, very little of this is done in the manuscript; more depth is needed on both parts. There is a considerable literature of Boolean network inference. The manuscript cites three references in which state or time course information was used to infer a Boolean model, but the context does not make it clear that these references also had algorithms (and in fact software). A comparison with existing methods is needed. Also, the discussion merely restates the sentence “our approach can be used to infer used to infer the mechanisms of signaling, gene-regulatory, and any other input-output processes in an automatic (i.e. fully unsupervised) way”; more discussion of the connection of a Boolean function to mechanism is needed.

More minor comments:

5. The “network” that is generated by this algorithm (see for example top right of Fig. 1) is in fact the state transition graph of the Boolean system. The nodes are states of the system and the edges are allowed transitions between states. The language needs to be clarified to avoid confusion with the interaction (or regulatory) network.

6. In line 14, “each edge corresponds to a Boolean rule or set of rules that describes the interaction between nodes “ is not sufficiently accurate and may be misunderstood. An edge between two gene products corresponds to an interaction. A Boolean rule describes the regulation of each node (gene product) by all of its regulators. It reflects all edges incident on a node. An edge can be equivalent to a Boolean rule if and only if there is a single regulator. An edge does not participate in multiple Boolean rules.

Reviewer #2: Let me start with my conclusions: the work presented by the authors is great, an interesting read, and a very good example of the much-needed work in formalizing Boolean model inference when applied to the dynamics of intracellular networks. I think this work would be a perfect addition to PLoS Computational Biology.

In this work, the authors describe a model inference framework for Boolean systems with stochastic asynchronous dynamics in which the required prior knowledge consists of mapping between initial conditions (IC) and attractors (A). They include several variants of their framework: one that includes only forward paths in state space from the IC to the attractor (F), one that includes backward transitions that still satisfy the IC->A conditions (B), one that includes expert knowledge in terms of prior-knowledge network (E), and others that constrain the model based on the frequency in which each attractor is reached using a genetic algorithm for model selection.

The authors introduce their method using the classic enzyme, substrate, product model of Michaelis-Menten kinetics, and then proceed to illustrate it to an 8-variable model of the Epithelial-to-Mesenchymal Transition. Some of their main findings are that (1) the forward-only variant often does not recapitulate many of the transitions observed in the reference model, (2) that backward-transition variant can often add too many more transitions than the ones in the reference model, (3) the expert knowledge approach can often help select the appropriate transitions and better match the reference model, and this is further improved when combined with the genetic algorithm approaches to match the frequency to reach each attractor, (4) some of the Boolean rules are not fully constrained by the IC->A + prior-knowledge network, but some are not.

I do have some comments and questions, which I think would greatly improve the manuscript if addressed:

Major:

1) “Boolean networks assume that the species are either present (1) or absent (0), i.e. E, S, ES, P \\in {0,1}, and that all reactions are equally likely, i.e. all rate constants are equal to 1.”

I do not agree with the “all reactions are equally likely” of this sentence, and I think this needs a clarification. What the authors describe is true for the case when the update probability of each node in the stochastic asynchronous scheme is set to be the same for all nodes. It is known that this not a good assumption for many systems, and some previous work has implemented more realistic alternatives (e.g., by updating different kind of nodes with different frequencies, see PMID: 22673395, PMID: 25189528).

I do think this is an important point that should be discussed (perhaps in a “Current limitations and future work” type of subsection?), since the frequency of reaching an attractor would be very sensitive to deviations from the uniform update probability of a node.

2) It is strange to me that the expert knowledge example of the Michaelis-Menten kinetics did not include the one based on the prior-network informed approach. There might be an underlying reason for this, but I think the authors should include this, as it becomes important in the more complicated EMT example.

3) Page 15. “I.e., if we want to eliminate a dependency of a variable, we need to eliminate each transition that does not include its symmetric counter part in the transition list.”

It is not clear to me that this the only way, but I do agree this is a possible way to “eliminate dependency of a variable”. E.g., if we have f(A,X,Y,Z,W), it is entirely possible that the real function is g(X,Y,A) = f(A=1,X,Y,Z=0,W=0), as long as it does not break the appropriate IC -> Attractor transitions. So, my general question is, (i) is it true that this way to eliminate dependency of a variable is able to obtain any function that can satisfy the IC->A priors, and (ii) is this true even if there is co-dependence between the functions of two variables (e.g. the IC->A priors can be satisfied if {f_variable1 = f1, f_variable2 = f2} or {f_variable1 = g1, f_variable2 = g2}, but not if {f_variable1 = f1, f_variable2 = g2} or {f_variable1 = g1, f_variable2 = f2}.

4) Page 16. “of 16 x 1 x 8388608 x 256 x 32 x 1 > 1e12 possible networks to choose from.” This is related to my above question on co-dependence between the functions of two variables that satisfy the IC->A priors. Shouldn't there be correlation between the choice of rules in one variable vs the one in the other? The fact that the number of rules is multiplied suggests that there is no correlation between the function in each of the variables. Is the multiplication of rule numbers possible because there is a guarantee that the IC -> Attractor transitions are respected?

5) On the inclusion of signs on the prior-knowledge network. Is there are a reason why the prior-knowledge network did not include the sign of the interactions? It seems to me that this might drastically reduce the number of possible Boolean functions in Table 7. Even if the algorithm has difficulty constraining Boolean functions to the sign-definite ones during the optimization processes, the authors should still be able to report how many of the ones in Table 7 are consistent with the known sign of the interactions (for the sign-definite functions of the reference model). If it is difficult to implement this, this should be discussed in a “Current limitations and future work” subsection.

6) Partial knowledge of the {IC->A} priors. I think it is important to have a discussion of what would happen if one only has partial knowledge of the {IC->A} priors, since this is one of the distinguishing features of this approach. Ideally, one would like a quantification for what happens with the two models described as one decreases from a full to partial knowledge of the {IC->A} priors. This is also important if one wants to apply this to a larger model, for which full knowledge of the {IC->A} priors would be impossible.

7) GitHub implementation. Currently, the GitHub implementation seems bare-bones. I would say that, at minimum, one should be able to reproduce the results of the manuscript, and it should be clear (through a detailed read me) how to do it. Ideally, one would like to be able to provide a set of {IC->A} priors, and possibly a network, and obtain the rules of the ensemble of models.

Others:

8) It is not clear to me if the forward-only network includes all possible orders of forward paths that can be obtained from the given ICs to the given attractors

9) Page 7. “Backward dynamic paths to enable dynamic loops (ES-B)”. I did not notice at first that ALL backward transitions are added (except the ones that destroy attractors). I was also very confused about how the information on which ones to add would be added, but I see the authors discuss that later in the section. I would recommend including a note on adding all allowed backward transitions being and oversimplification, and that this will be discussed later.

Minor:

- Figures S1-S2 are very hard to follow as they currently are. Additional explanation of the underlying “notation” of the figure and the colors would be helpful.

- Figure S3-S6. A legend for the color scheme would be extremely helpful, so that one doesn’t have to rely on the caption. Also a more detailed description of the figure would be helpful (e.g., red are the target attractors).

- The authors use the word “pathway” to describe paths in the state transition network. I have to say I was first confused by this terminology, since “pathway” has a particular biological meaning that is associated with the biological network, while the authors use it to describe the state transition network. I would recommend using “path” or “path towards X”, or some other alternative, instead of “pathway”.

- Fig 3. “The darker the arrow, the more simulation steps are necessary to reach this particular transition in principle.” I am not sure what the authors mean with “in principle”. I think that extra pair of words is not needed.

- Fig 3. “The x-axis represents the number of simulation steps it takes to reach the steady state.”. I am not sure the “it takes to reach the steady state” part is needed.

- Page 10. “Application to an established model: Epithelial to Mesenchymal Transitios”. There is a typo; this should say “Transition”.

- Page 12. “we extended Table 4 by the according frequencies for our automatically created”. There is a typo, “Table 4 by the according frequencies”.

- Page 10. “For the other three subnetworks, the frequencies to enter into a given steady state generally match well with EMT-O.”. I am not sure I agree. (1,1) does seem to match, (0,1) must match (by construction), but (0,0) seems slightly better than the (0,1) results but not by much.

Reviewer #3: Uploaded as attachment.

**Have all data underlying the figures and results presented in the manuscript been provided?**

Reviewer #1: Yes

Reviewer #2: Yes

Reviewer #3: None

PLOS authors have the option to publish the peer review history of their article (what does this mean?). If published, this will include your full peer review and any attached files.

Reviewer #1: No

Reviewer #2: No

Reviewer #3: No
---

## [Decision Letter · Decision Letter 1]

3 May 2021

Dear Dr. Lopez,

We are pleased to inform you that your manuscript 'Unsupervised logic-based mechanism inference for network-driven biological processes' has been provisionally accepted for publication in PLOS Computational Biology.

Best regards,

Jeffrey J. Saucerman

Associate Editor

PLOS Computational Biology

Jason Haugh

Deputy Editor

PLOS Computational Biology

Reviewer's Responses to Questions

**Comments to the Authors:**

Reviewer #1: The revised manuscript is much improved and successfully addresses my points.

Reviewer #2: The authors have addressed my concerns with their revisions. I am happy to recommend acceptance of their article.

Reviewer #3: The authors have addressed all my concerns.

**Have the authors made all data and (if applicable) computational code underlying the findings in their manuscript fully available?**

Reviewer #1: None

Reviewer #2: Yes

Reviewer #3: Yes

PLOS authors have the option to publish the peer review history of their article (what does this mean?). If published, this will include your full peer review and any attached files.

Reviewer #1: No

Reviewer #2: **Yes: **Jorge Gomez Tejeda Zanudo

Reviewer #3: No

---

## [Editor Report · Acceptance letter]

18 May 2021

PCOMPBIOL-D-20-02285R1 

Unsupervised logic-based mechanism inference for network-driven biological processes

Dear Dr Lopez,

I am pleased to inform you that your manuscript has been formally accepted for publication in PLOS Computational Biology. Your manuscript is now with our production department and you will be notified of the publication date in due course.

With kind regards,

Zsofi Zombor
